# Arabidopsis TRB proteins function in H3K4me3 demethylation by recruiting JMJ14

Ming Wang[1,7], Zhenhui Zhong[1,7], Javier Gallego-Bartolomé[1,6], Suhua Feng[1,2], Yuan-Hsin Shih[1,3], Mukun Liu[1], Jessica Zhou[1], John Curtis Richey[1], Charmaine Ng[1], Yasaman Jami-Alahmadi[4], James Wohlschlegel[4], Keqiang Wu[3] & Steven E. Jacobsen[1,2,4,5] ✉

Arabidopsis telomeric repeat binding factors (TRBs) can bind telomeric DNA sequences to protect telomeres from degradation. TRBs can also recruit Polycomb Repressive Complex 2 (PRC2) to deposit tri-methylation of H3 lysine 27 (H3K27me3) over certain target loci. Here, we demonstrate that TRBs also associate and colocalize with JUMONJI14 (JMJ14) and trigger H3K4me3 demethylation at some loci. The *trb1/2/3* triple mutant and the *jmj14-1* mutant show an increased level of H3K4me3 over TRB and JMJ14 binding sites, resulting in up-regulation of their target genes. Furthermore, tethering TRBs to the promoter region of genes with an artificial zinc finger (TRB-ZF) successfully triggers target gene silencing, as well as H3K27me3 deposition, and H3K4me3 removal. Interestingly, JMJ14 is predominantly recruited to ZF off-target sites with low levels of H3K4me3, which is accompanied with TRB-ZFs triggered H3K4me3 removal at these loci. These results suggest that TRB proteins coordinate PRC2 and JMJ14 activities to repress target genes via H3K27me3 deposition and H3K4me3 removal.

Arabidopsis Telomere Repeat Binding factors (TRBs) are well known for their role in the maintenance of chromosome ends through binding to the telomeric repeat DNA sequences[1–4]. They can protect the telomeres from fusion and degradation, as well as from being misrecognized as unpaired chromosome breaks[1–4]. Arabidopsis has three TRB proteins with a clear coiled-coil domain, TRB1, 2, and 3, which contain a single Myb-like domain followed by a histone-like domain, GH1/GH5, and a C-terminal coil-coiled domain. The Myb-like domain is required for DNA binding[4,5], while the GH1/GH5 histone-like domain is involved in protein-protein interactions, including the interactions between the three TRBs themselves[6].

In addition to the protection of telomeres, TRB proteins are also found in the PWWPs-EPCRs-ARIDs-TRBs (PEAT) complex, which is involved in heterochromatin condensation and silencing likely via histone deacetylation[7]. Mutation of *TRB1* and *TRB3* causes enhanced phenotypic defects in the *LIKE HETEROCHROMATIN PROTEIN1* (*lhp1*) mutant[8], which is a Polycomb Repressive Complex 2 (PRC2) associated protein and a reader of histone tri-methylation H3K27 (H3K27me3)[9,10]. TRB1, 2, and 3 are functionally overlapping, such that the *trb* single or double mutants fail to produce any morphological phenotype[5], while the *trb1/2/3* triple mutant shows a similar developmental phenotype and transcriptomic profiles as mutations in the components of PRC2, including *SWINGER* (*swn*) and *CURLY LEAF* (*clf*) mutants, which are the key enzymes for H3K27me3 deposition[5]. Interaction of TRBs with SWN and CLF, and loss of H3K27me3 in *trb1/2/3* triple mutant over the TRB binding motifs, such as telobox related motifs, suggest that TRBs

[1]Department of Molecular, Cell and Developmental Biology, University of California at Los Angeles, Los Angeles, CA 90095, USA. [2]Eli & Edythe Broad Center of Regenerative Medicine & Stem Cell Research, University of California at Los Angeles, Los Angeles, CA 90095, USA. [3]Institute of Plant Biology, National Taiwan University, Taipei 10617, Taiwan. [4]Department of Biological Chemistry, University of California at Los Angeles, Los Angeles, CA 90095, USA. [5]Howard Hughes Medical Institute (HHMI), University of California at Los Angeles, Los Angeles, CA 90095, USA. [6]Present address: Instituto de Biología Molecular y Celular de Plantas (IBMCP), CSIC-Universitat Politècnica de València, 46022 Valencia, Spain. [7]These authors contributed equally: Ming Wang, Zhenhui Zhong. ✉e-mail: jacobsen@ucla.edu

recruit the PRC2 complex to certain target sites for H3K27me3 deposition[5].

The H3K27me3 histone mark is associated with gene repression, while H3K4me3 is associated with active gene expression. In mammalian stem cells, many genes are associated with both H3K27me3 and H3K4me3, a state called bivalency[11,12]. In plants, however, H3K27me3 and H3K4me3 are almost entirely mutually exclusive[13]. The mechanism by which these two marks are partitioned into non-overlapping genomic domains is poorly understood. While Polycomb group (PcG) proteins are responsible for H3K27me3 deposition, trithorax group (trxG) proteins mediate H3K4 deposition[14]. In plants, the PRC2 complex can be recruited by VIVIPAROUS1/ABI3-LIKE1/2 (VAL1/2), BASIC PENTACYSTEINE 1 (BPC1), and ARABIDOPSIS ZINC FINGER1 (AZF1), in addition to TRBs[15–17]. The removal of H3K4me3 is controlled by H3K4 demethylases, which include Jumonji-domain containing proteins (JMJ14-18)[18–22], and Lysine-Specific Demethylase 1 Like proteins, LDL1-3 and FLD (FLOWERING LOCUS D)[23–25]. JMJ14 is a well-studied H3K4 demethylase that removes H3K4 di- and tri-methylation and forms a complex with two NAC type transcription factors, NAC050 and NAC052[18,26–28].

In this study, we show that TRBs interact with JMJ14 as well as its associated NAC050 and NAC052 proteins. We demonstrate that TRB proteins not only recruit PRC2 complexes to deposit H3K27me3[5,8], but also recruit JMJ14 to remove H3K4me3, which can partially explain the mutual antagonism of these two histone marks over certain regions that are co-bound by JMJ14 and TRBs. RNA-seq results of a *trb1/2/3* triple mutant also revealed that the up-regulated genes largely overlapped with those in the *jmj14-1* mutant. In addition, H3K4me3 was enriched in both *jmj14-1* and *trb1/2/3* mutants over JMJ14 and TRB binding sites. Finally, we fused TRBs with a zinc finger (TRB-ZF) targeting the Arabidopsis *FWA* gene and showed that it successfully repressed *FWA* expression as well as many other genes that were bound by this zinc finger, and this repression was associated with a combination of H3K4me3 demethylation and H3K27me3 deposition. Interestingly, the removal of H3K4me3 in TRB-ZF lines mainly occurred over the ZF off-target sites with medium or low levels of pre-existing H3K4me3, which is consistent with chromatin immunoprecipitation sequencing (ChIP-seq) results showing that JMJ14 was mainly recruited to these regions by TRB-ZFs. This is further consistent with the endogenous JMJ14 binding loci showing low levels of H3K4me3, suggesting that there might be certain unknown mechanisms that prohibit JMJ14 from accessing H3K4me3 enriched loci. Together these results suggest that the TRBs act via the recruitment of H3K27me3 methylation and H3K4me3 demethylation activities to enforce gene silencing.

## Results

### TRB1/2/3 (TRBs) interact and partially colocalize with JMJ14

To uncover TRB interacting proteins, we generated pTRB:TRB-FLAG (FLAG-TRB) transgenes in their respective mutant backgrounds and performed native and cross-linked immunoprecipitation mass spectrometry (IP-MS). Consistent with previous studies, TRBs pulled down LHP1 and several PRC2 subunits[5], the PRC2 accessory protein ICU11[29], and the PEAT complex[7] (Supplementary Data 1). Interestingly, peptides of the Arabidopsis histone H3K4me3 demethylase JMJ14 as well as its two NAC domain interactors, NAC050 and NAC052[27,28], were also pulled down by FLAG-TRBs (Supplementary Data 1), suggesting that TRBs may recruit the JMJ14 to remove H3K4me3, thus contributing to gene silencing.

To further confirm the interaction, we performed co-immunoprecipitation experiments by crossing pJMJ14:JMJ14-Myc (Myc-JMJ14) with FLAG-TRB1, FLAG-TRB2, and FLAG-TRB3, respectively. Consistent with our IP-MS results, JMJ14 was also pulled down by all three TRBs (Fig. 1a and Supplementary Fig. 1a). We also fused JMJ14 and NAC052 with the N-terminal fragment of YFP, and TRBs with the C-terminal fragment of YFP to perform bimolecular fluorescence

complementation (BiFC) in *Nicotiana benthamiana*. YFP signals were observed when TRBs were co-expressed with JMJ14 or NAC052, but not with the empty vector (EV) control (Supplementary Fig. 1b, c), suggesting that TRBs interact with both JMJ14 and NAC052.

To study the natural function of TRBs and JMJ14, we performed ChIP-seq utilizing FLAG-TRBs and FLAG-JMJ14 transgenic lines to examine target sites throughout the genome. We also included published ChIP-seq data of HA-NAC050 and HA-NAC052 in the analysis[28]. We found that TRB1, TRB2, and TRB3 were highly co-localized throughout the genome (Fig. 1b and Supplementary Fig. 2a). Consistent with the observed TRBs-JMJ14 and TRBs-NAC050/052 interaction, we also observed an overlap between JMJ14, TRBs, and NAC050/052 peaks (Fig. 1b, c, and Supplementary Fig. 2b, c). TRBs colocalized with majority of the JMJ14 sites (Fig. 1c), and both TRB1 and JMJ14 showed strong signals over most of HA-NAC050 peaks (Supplementary Fig. 2b). However, there were still a large fraction of TRB1 peaks that did not overlap with JMJ14 peaks (Fig. 2a). We therefore divided the TRB1 peaks into two clusters: peaks that overlapped JMJ14 peaks (Cluster 1) and peaks that did not (Cluster 2) (Fig. 2a). Next, we examined the enrichment of H3K4me3 ChIP-seq signals within the two clusters of TRB1 peaks. TRB1 Cluster 1 peaks that also colocalized with JMJ14 showed very low levels of H3K4me3, consistent with the demethylation activity of JMJ14 over its binding sites (Fig. 2b, c). Cluster 2 peaks showed much higher levels of H3K4me3, especially at the flanks of TRB peak centers (Fig. 2b, c). This is likely because TRB peaks were most often in promoter regions while H3K4me3 is usually located after the transcription start sites (TSS) and in the 5′ transcribed regions of genes (Fig. 2b, c). Consistently, NAC050 and NAC052 only co-localized with TRB1 Cluster 1 peaks, but not Cluster 2 peaks (Fig. 2c and Supplementary Fig. 2c), suggesting that TRBs might form a complex with JMJ14 and NAC050/052 over their co-bound regions.

Interestingly, 79% of the peaks in Cluster 2 were located at promoter and 5′ UTR regions (Fig. 2d), which is consistent with TRBs' role as Myb-type transcription factors[30]. In comparison, less than 60% of the peaks in Cluster 1 were located at promoter and 5′ UTR regions (Fig. 2d). Instead, 25% of TRB1 Cluster 1 peaks were located in exons compared to less than 5% of those in Cluster 2 (Fig. 2d). To further track the distribution patterns of JMJ14 and TRB1 peaks over their binding genes, we plotted the FLAG-JMJ14 and FLAG-TRB1 ChIP-seq signals over the genes proximal to Cluster 1 and 2. As expected, JMJ14 ChIP-seq signals were distributed over gene body regions of Cluster 1 proximal genes (Supplementary Fig. 3a). In addition, Cluster 1 proximal genes showed higher FLAG-TRB1 ChIP-seq signals over gene body regions, as well as lower ChIP-seq signals over TSS regions, as compared to Cluster 2 proximal genes (Supplementary Fig. 3b). Also consistent with the JMJ14 localization, H3K4me3 ChIP-seq signals were much lower in Cluster 1 genes than in Cluster 2 genes (Supplementary Fig. 3c). These data show that the colocalization of TRBs and JMJ14 occurs most often in gene bodies and corresponds to low levels of H3K4me3.

Previous work has shown that the peaks of TRB1 ChIP-seq are enriched in the telobox motif[8]. Consistent with this result, our motif prediction using TRB binding sites showed an enrichment for sequences similar to the Arabidopsis telomere repeat sequence TTAGGG, not only in TRB1 ChIP-seq, but also in TRB2 and TRB3 ChIP-seq datasets (Supplementary Fig. 4a). In addition, motif prediction of JMJ14 and NAC050/052[28] peaks showed a strong enrichment of the CTTGnnnnnCAAG motif (Supplementary Fig. 4b), which is consistent with previous findings that both JMJ14 and NAC050/NAC052 bind this motif[28]. Interestingly, both TTAGGG and CTTGnnnnnCAAG sequences were enriched in the TRBs, JMJ14, and NAC050/052 ChIP-seq peaks (Supplementary Fig. 4a, b). However, the top predicted motif bound by TRBs was TTAGGG, with the CTTGnnnnnGAAAG motif ranking number 61 (TRB1), 82 (TRB2), or 69 (TRB3). On the other hand, the predicted motifs for JMJ14, NAC050, and NAC052 bound sites were

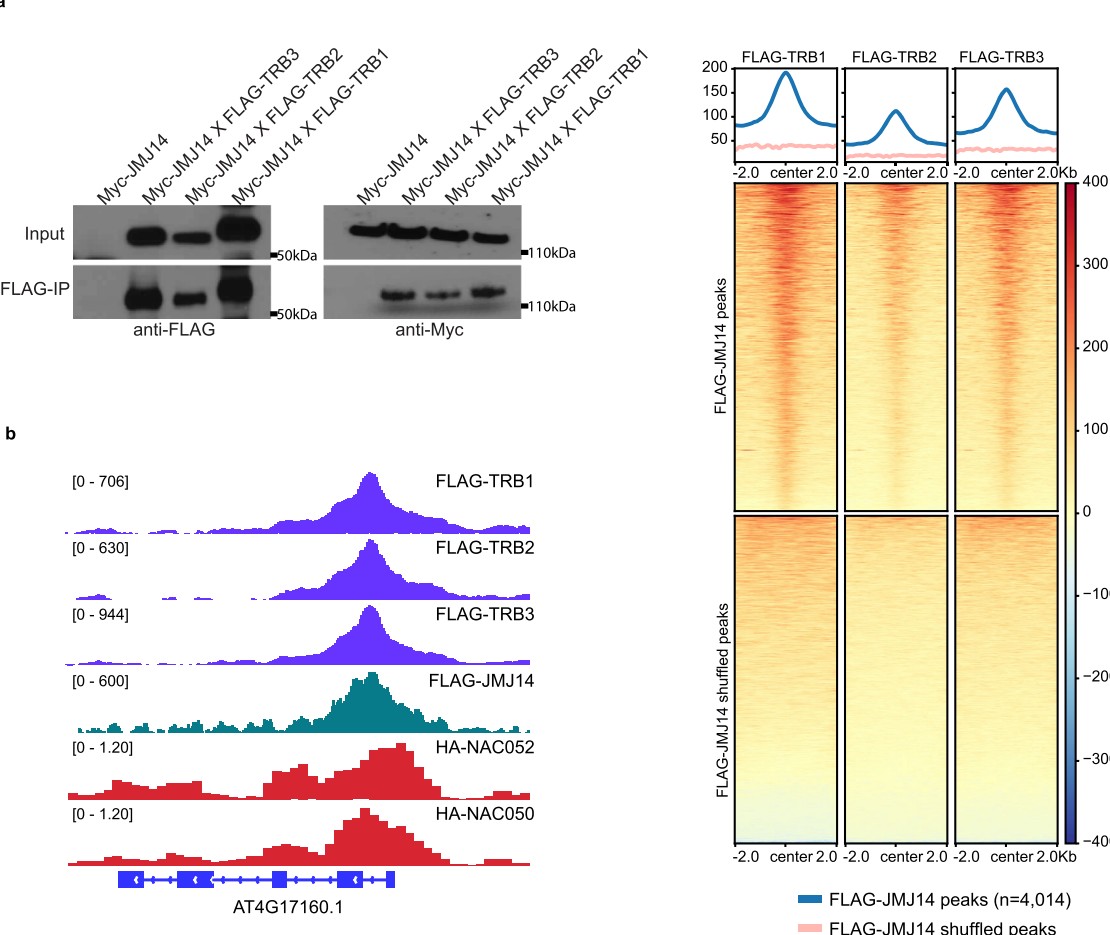

**Fig. 1 | Arabidopsis TRBs interact with JMJ14. a** Western blot showing a Co-immunoprecipitation (Co-IP) assay in Myc-JMJ14 F2 crossed lines with FLAG-TRB3, FLAG-TRB2, and FLAG-TRB1, respectively. Similar results were observed in two biological replicates. **b** Screenshots showing normalized ChIP-seq signals in FLAG-TRB1, FLAG-TRB2, FLAG-TRB3, FLAG-JMJ14, HA-NAC052, and HA-NAC050 over a representative co-bound gene. **c** Metaplots and heatmaps depicting the ChIP-seq signals of FLAG-TRB1, FLAG-TRB2, and FLAG-TRB3 over FLAG-JMJ14 peaks and shuffled peaks (*n* = 4014).

CTTGnnnnnCAAG at rank number 1, and TTAGGG at number 42 (JMJ14), 42 (NAC050), or 41 (NAC052) (Supplementary Fig. 4a, b). These results suggest JMJ14 may be independently recruited to telebox sites by TRBs and to CTTGnnnnnCAAG sites by NAC050/052.

We divided the TRB1 and JMJ14 co-bound regions into four clusters based on whether they contained the Telobox binding sequence TTAGGG (37% of regions), the NAC binding sequence CTTGnnnnnCAAG (14% of regions), or both sequences (12% of regions), or neither sequence (37% of regions) (Supplementary Fig. 4c). Compared to shuffled peaks, these TRB/JMJ14 co-bound sites showed a strong enrichment for the CTTGnnnnnCAAG sequence (Supplemental Fig. 4c). We found that the TRB ChIP-seq signals were higher at regions with the TTAGGG only sequence, or with both sequences, than at sites with only the CTTGnnnnnGAAG sequence or neither sequence, while JMJ14 and the NACs bound more to the regions with only CTTGnnnnnGAAG or with both sequences, than to those with TTAGGG only or neither sequence (Supplementary Fig. 4d). These results again support the conclusion that JMJ14 is likely recruited to both the TRB binding sites and the NAC050/052 binding sites independently.

**TRBs and JMJ14 contribute to H3K4me3 demethylation in vivo**
*TRB1*, *TRB2*, and *TRB3* are redundant homologs, and the single or double mutants of any combination show no morphological phenotype[5]. A *trb1/2/3* triple mutant allele was reported with a strong dwarf phenotypic defect[5]. We utilized different T-DNA insertion

mutant lines to create a second *trb1/2/3* triple mutant allele[31], which also exhibited a similar morphological defect (Supplementary Fig. 5)[5]. *trb1/2/3* seedlings were very small when grown on solid media, and when transplanted onto soil, they often failed to survive and were completely infertile (Supplementary Fig. 5).

In order to investigate the role of TRBs in regulating H3K27me3, we performed ChIP-seq with anti-H3K27me3 in *trb1/2/3* triple mutant and Col-0 wild type plants. We observed 7975 decreased and 2184 increased H3K27me3 regions in *trb1/2/3* (Supplementary Fig. 6a, b, and Supplementary Data 2). The H3K27me3 levels were also strongly reduced over TRB1, 2, and 3 binding sites in the *trb1/2/3* triple mutant when compared to Col-0 wild type (Supplementary Fig. 6b, c).

To investigate the function of TRBs and JMJ14 in H3K4me3 regulation, we also performed H3K4me3 ChIP-seq in *trb1/2/3* and *jmj14-1* mutants and found that H3K4me3 levels were strongly increased at JMJ14 binding sites in both the *trb1/2/3* triple mutant and the *jmj14-1* mutant (Fig. 3a, b). Interestingly, H3K4me3 ChIP-seq signals were also increased in the *trb1/2/3* triple mutant at TRB1 Cluster 2 peaks, which did not overlap with JMJ14 peaks (Fig. 3b–d), suggesting that TRBs likely influence the level of H3K4me3 in both JMJ14-dependent and -independent manners. It seemed possible this could be due to the loss of H3K27me3 in *trb1/2/3*, especially because H3K4me3 and H3K27me3 have been shown to be in some cases mutually antagonistic[32–35]. However, we observed a near perfect opposite trend of changes of H3K4me3 and H3K27me3 in the *trb1/2/3* mutant versus Col-0 at the

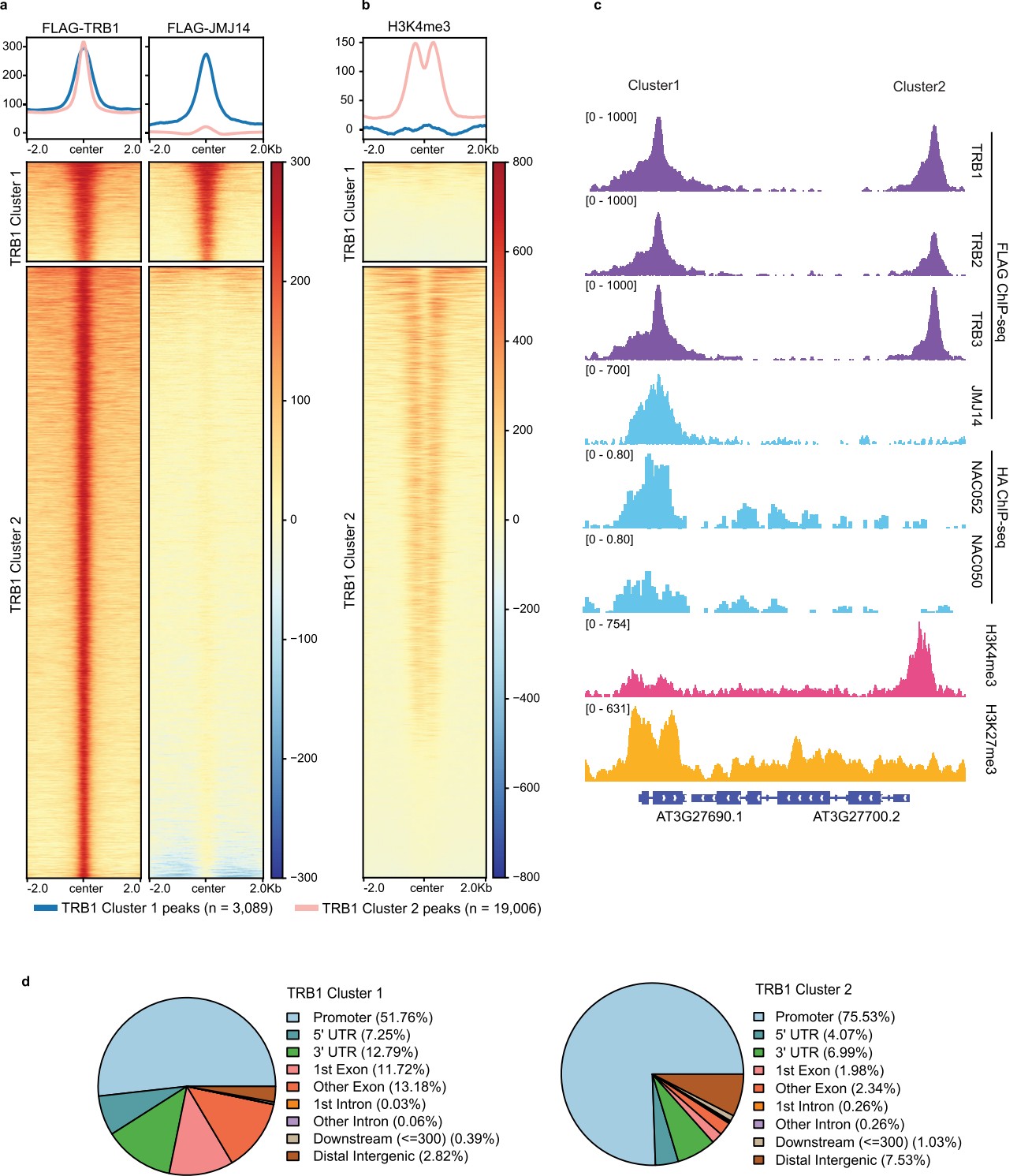

**Fig. 2 | TRB1 peaks overlap with JMJ14 peaks at H3K4me3 depleted regions.**
**a**, **b** Heatmaps and metaplots showing ChIP-seq signals of FLAG-TRB1 (**a**, left panel), FLAG-JMJ14 (**a**, right panel), and H3K4me3 (**b**) over TRB1 Cluster 1 peaks (*n* = 3089) and Cluster 2 peaks (*n* = 19,006), respectively. **c** Screenshots displaying the ChIP- seq signals of FLAG-TRB1, FLAG-TRB2, FLAG-TRB3, FLAG-JMJ14, HA-NAC052, HA-NAC050, H3K4me3, and H3K27me3 over a representative locus containing TRB1 Cluster 1 and Cluster 2 peaks. **d** Pie Charts depicting the annotations of the TRB1 Cluster 1 peaks (left panel) and Cluster 2 peaks (right panel).

TRB1 Cluster 1 peaks, we did not observe this at Cluster 2 peaks (Fig. 3b and Supplementary Fig. 7a, b). Moreover, we found a total of 2617/3736 (70%) of JMJ14 bound genes overlapped with 2617/14,882 (18%) of TRB bound genes (Supplementary Data 3), among which 860 (33%) showed a switch from H3K27me3 to H3K4me3 in the *trb1/2/3* mutant (Sup- plementary Data 4), suggesting that TRB1 and JMJ14 can trigger a shift

from H3K4me3 to H3K27me3 in about a third of TRB1 and JMJ14 co- bound genes. We found that there were a much higher number of regions with increased H3K4me3 (16,634) than regions with decreased H3K27me3 (7,975) in the *trb1/2/3* mutant (Supplementary Data 2). In addition, the majority of H3K27me3 reduced regions in *trb1/2/3* did not gain H3K4me3 ChIP-seq signals (Supplementary Fig. 8a, b). These data

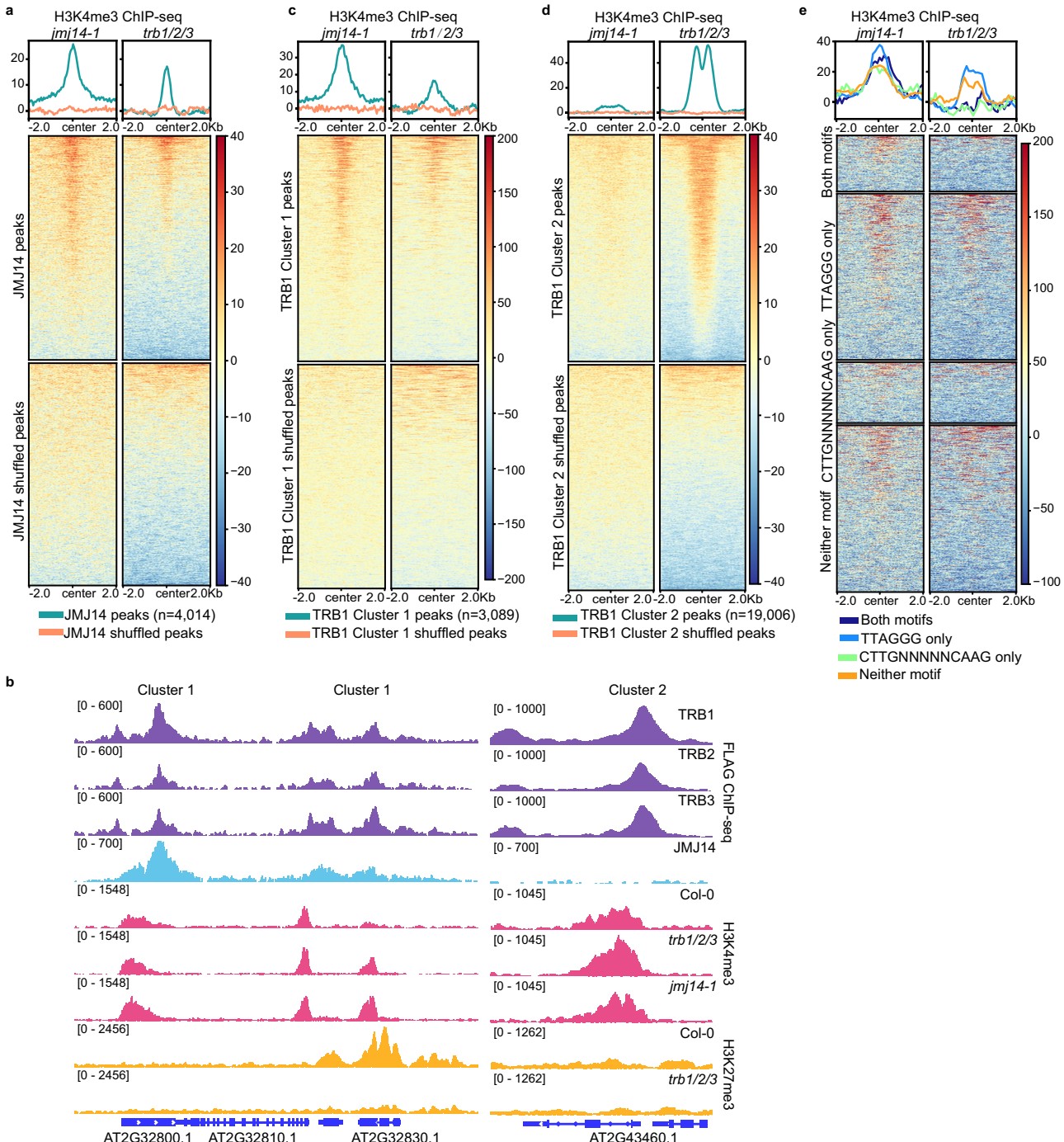

**Fig. 3 | H3K4me3 ChIP-seq signals are increased in both *jmj14-1* and *trb1/2/3* mutants. a** Metaplots and heatmaps showing the normalized H3K4me3 ChIP-seq levels of *jmj14-1* and *trb1/2/3* mutants versus Col-0 wild type plants, over FLAG-JMJ14 peaks and shuffled peaks (*n* = 4014). **b** Screenshots displaying the FLAG ChIP-seq signals of FLAG-TRB1, FLAG-TRB2, FLAG-TRB3, and FLAG-JMJ14, histone H3K4me3 ChIP-seq signals of Col-0, *trb1/2/3* and *jmj14-1* mutants, and H3K27me3 ChIP-seq signals of Col-0 and *trb1/2/3* mutant over four representative loci. **c**–**e** Metaplots and heatmaps showing H3K4me3 ChIP-seq signals of *jmj14-1* and *trb1/2/3* mutants versus Col-0 wild type plants over TRB1 Cluster 1 peaks and shuffled peaks (**c**, *n* = 3089), TRB1 Cluster 2 peaks and shuffled peaks (**d**, *n* = 19,006), and TRB1 JMJ14 co-bound peaks displaying the TTAGGG motif only (*n* = 1176), the CTTGnnnnnCAAG motif only (*n* = 431), both motifs (*n* = 396), and neither (*n* = 1180), respectively (**e**).

suggest that the induction of H3K4me3 in *trb1/2/3* mutant can be explained only partially by changes in H3K27me3 levels.

We also plotted the changes of H3K4me3 in the *trb1/2/3* triple mutant or the *jmj14-1* mutant over the four clusters of TRB/JMJ14 co-bound sites described above (Supplementary Fig. 4 c, d). In the *trb1/2/3* triple mutant, H3K4me3 was gained over the two clusters of regions that are lacking a CTTGnnnnnGAAG sequence (the Telobox only cluster and the neither motif cluster) but not at the two clusters that do have the CTTGnnnnnGAAG sequence (Fig. 3e). However, all four clusters of sites gained H3K4me3 in *jmj14-1* (Fig. 3e). This result suggests that JMJ14-NAC050/052 likely can act at the CTTGnnnnnGAAG containing regions independently of TRBs, again suggesting that JMJ14 is likely recruited to both the TRB binding sites and the NAC050/052 binding sites independently.

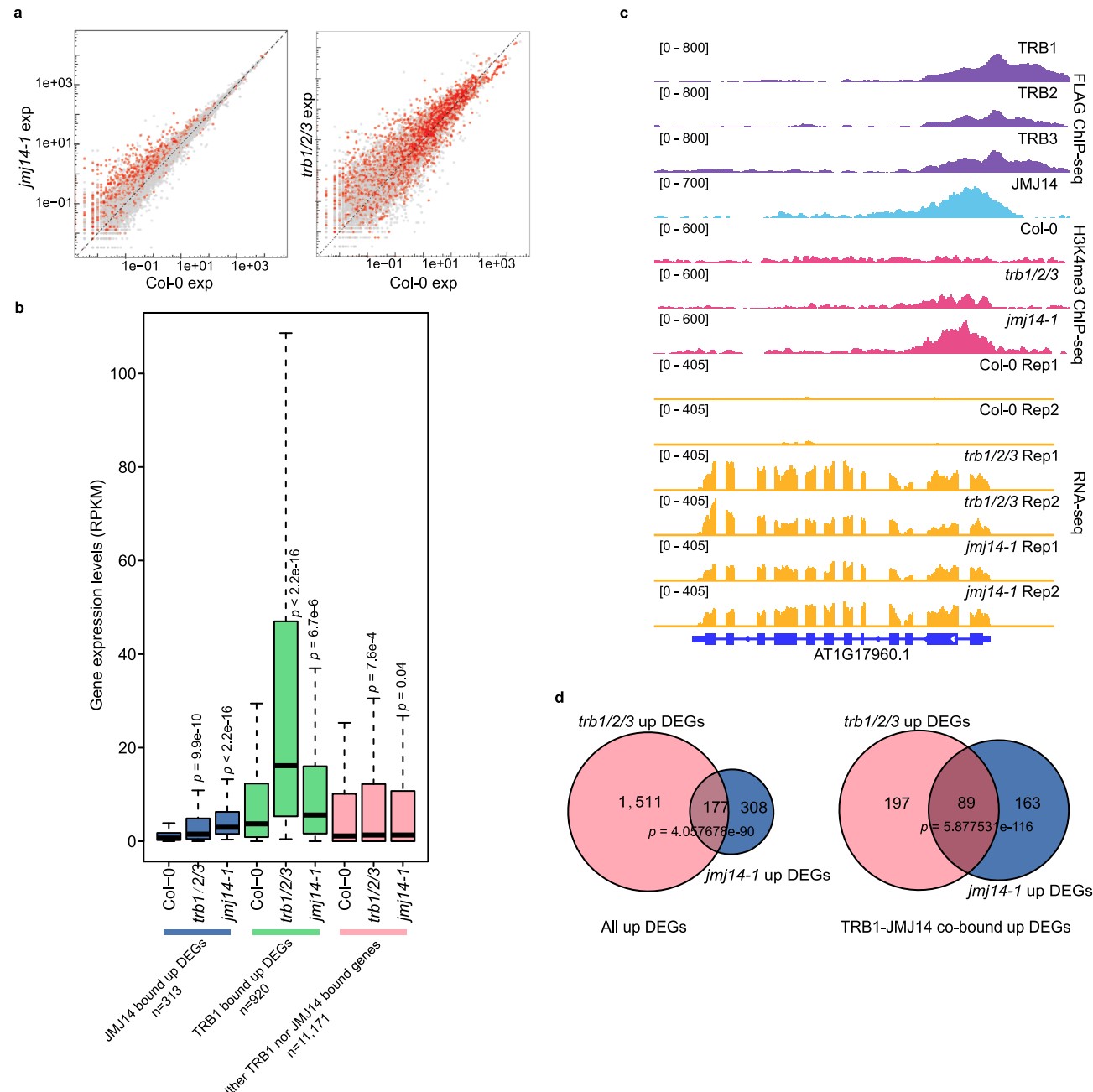

**Fig. 4 | Up-regulated DEGs in *jmj14-1* and *trb1/2/3* mutants are partially overlapping. a** Scatterplot showing the gene expression level in *jmj14-1* vs Col-0 (left panel) and *trb1/2/3* vs Col-0 (right panel). The red dots highlight genes bound by JMJ14 (left panel) or TRB1 (right panel). **b** Box plots depicting the expression level of FLAG-JMJ14 bound and up-regulated DEGs, FLAG-TRB1 bound and up-regulated DEGs, and neither TRB1 nor JMJ14 bound genes, in Col-0, *trb1/2/3*, and *jmj14-1* mutants, respectively (*n* = 3 biologically independent samples). The top, mid-line, and bottom of the boxplots represent the upper quartile, median, and lower quartile, respectively. The whiskers represent the minimum and maximum values of the dataset. The *p* value is estimated by two-sided *t*-test. **c** Screenshots describing the FLAG ChIP-seq signals of FLAG-TRB1, FLAG-TRB2, FLAG-TRB3, and FLAG-JMJ14, H3K4me3 ChIP-seq signals of Col-0, *trb1/2/3*, and *jmj14-1* mutants, as well as the RNA-seq signals of Col-0, *trb1/2/3*, and *jmj14-1* mutants. **d** Venn diagrams showing the overlap of all the up-regulated DEGs (left panel) or JMJ14-TRB1 co-bound up-regulated DEGs (right panel) between the *trb1/2/3* triple mutant and the *jmj14-1* mutant. The *p* value is estimated by one-sided hypergeometric test.

## TRBs and JMJ14 silence a common set of target genes

To uncover genes that are regulated by TRBs and JMJ14, we performed RNA-seq in the *trb1/2/3* triple mutant and compared it to *jmj14-1* mutant and Col-0 wild type. Both mutants exhibited many up-regulated DEGs (Fig. 4a, Supplementary Fig. 9a, and Supplementary Data 5), and JMJ14 and TRB binding genes showed an even higher density of up-regulated DEGs in *jmj14-1* and *trb1/2/3* mutants, respectively (Fig. 4a-c and Supplementary Fig. 9b). Moreover, up-regulated DEGs in both mutants showed an increased level of

H3K4me3, when compared with wild type plants (Fig. 4c and Supplementary Fig. 9c), suggesting that JMJ14 and TRBs direct H3K4me3 demethylation and cause gene silencing. If TRBs and JMJ14 coordinately regulate gene expression, we would expect a common set of DEGs between *trb1/2/3* and *jmj14-1* mutants. Indeed, we found the DEGs of the *trb1/2/3* triple mutant and the *jmj14-1* mutant significantly overlapped with a total of 177 out of 485 up-regulated DEGs in the *jmj14-1* mutant also up-regulated in the *trb1/2/3* triple mutant (Fig. 4d and Supplementary Data 5). This overlap was

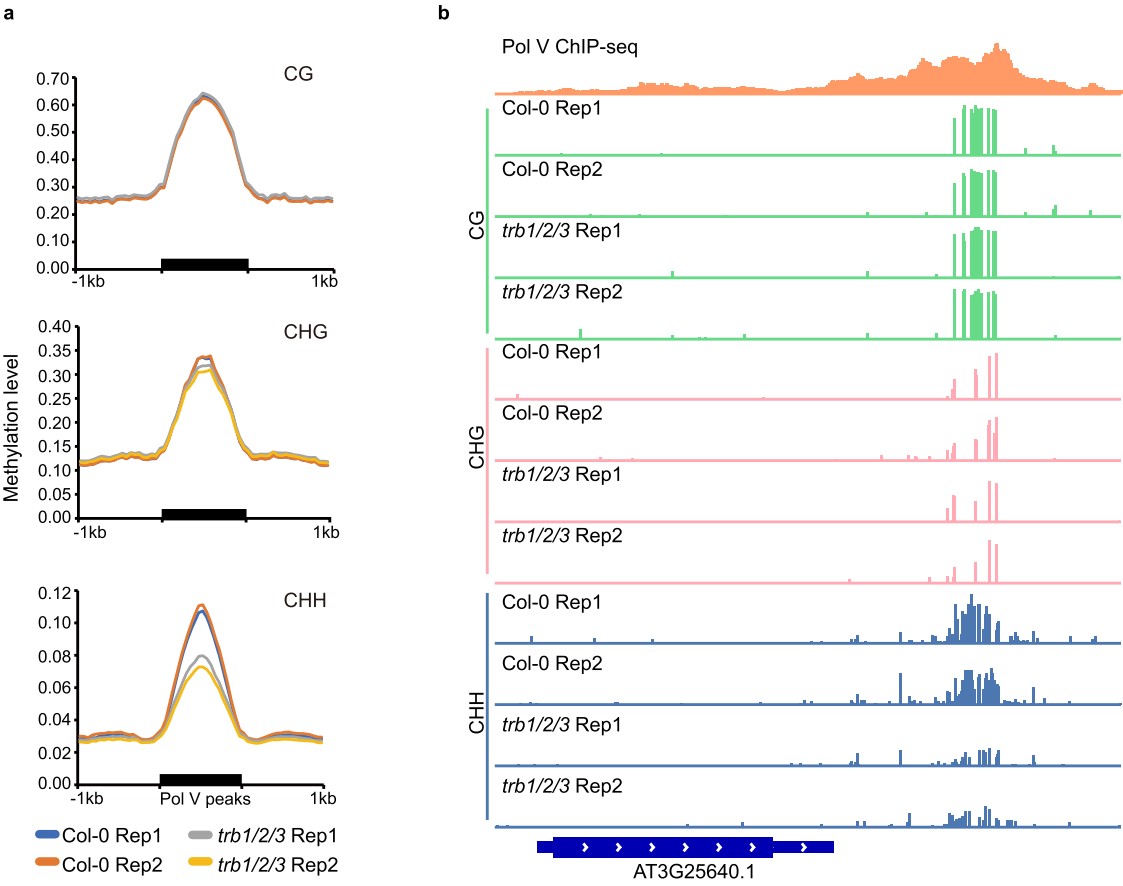

**Fig. 5 | Non-CG DNA methylation is reduced in the *trb1/2/3* mutant over RdDM sites. a** CG, CHG, and CHH DNA methylation level in two replicates of Col-0 and *trb1/2/3* triple mutants over RdDM sites, measured by whole genome bisulfite sequencing (WGBS). **b** Screenshots of CG, CHG, and CHH DNA methylation level in Col-0 and *trb1/2/3* mutants, respectively, over a representative RdDM site.

even greater when only including genes that were bound by both TRB1 and JMJ14 (Fig. 4d).

## Non-CG DNA methylation was reduced in the *trb1/2/3* triple mutant

Previous reports showed that the *jmj14-1* mutant exhibited a mild reduction in CHH DNA methylation at sites of RNA-directed DNA methylation[36,37]. Because of the interaction between TRBs and JMJ14, we therefore performed WGBS in the *trb1/2/3* triple mutant. Indeed, we observed a strong reduction of CHH DNA methylation, and a moderate loss of CHG DNA methylation at RdDM sites in *trb1/2/3* (Fig. 5a, b). We also found that CHH DNA methylation was reduced in the *trb1/2/3* mutant in the 5′ and 3′ ends of genes containing TRB1 and JMJ14 peaks (Supplementary Fig. 10a–c), which is likely because RdDM sites tend to be at the flanks of genes.

## TRBs trigger target gene silencing when fused with an artificial zinc finger

To further characterize the function of TRB proteins, we used a gain of function approach by tethering TRB proteins with the artificial zinc finger 108 (ZF), which was designed to target the promoter region of the Arabidopsis *FWA* gene. These TRB-ZF fusions were individually transformed into the Arabidopsis *fwa* epiallele, in which DNA methylation at the *FWA* promoter region has been inheritably and permanently lost, leading to *FWA* over-expression and a delayed flowering phenotype[38]. We found that TRB2-ZF and TRB3-ZF, and to a lesser extent TRB1-ZF (collectively referred to as TRB-ZFs in the subsequent text) caused an early flowering phenotype in the *fwa* epiallele background (Fig. 6a and Supplementary Fig. 11a), by silencing *FWA*

expression (Fig. 6b). There was a wide variation in the flowering time effect in different transgenic lines, and we found that the lines showing early flowering had stronger protein expression than lines showing no effect (Supplementary Fig. 11b). We also performed bisulfite PCR (BS-PCR) to examine whether DNA methylation was restored at the *FWA* promoter region, however DNA methylation was unaltered (Supplementary Fig. 12a), suggesting that TRB-ZFs triggered *FWA* gene silencing independent of DNA methylation. In line with a DNA methylation independent mechanism, the TRB-ZF fusion induced early flowering phenotype was not heritable in T2 lines that had segregated away the transgenes (Supplementary Fig. 12b).

Our previous studies have shown that ZF108 has thousands of off-target binding sites throughout the Arabidopsis genome, in addition to the *FWA* promoter region[39,40]. Therefore, we exploited these off-target regions to determine if TRB-ZFs could also silence genes near these ZF off target sites. Towards this end, we performed RNA-seq on TRB-ZFs T2 transgenic lines showing an early flowering phenotype and compared this with an *fwa* control plant. We performed Region Associated DEG (RAD) analysis using the differentially expressed genes (DEGs) in TRB-ZF lines[41], which revealed a significantly higher number of down-regulated DEGs than up-regulated DEGs when the ZF peaks were within one kilobase of the TSS of the genes (Fig. 6c and Supplementary Fig. 13). Thus TRB-ZFs can silence many other genes in addition to *FWA*.

To test whether TRB-ZF mediated target gene silencing is caused by the deposition of H3K27me3, the demethylation of H3K4me3, or both, we performed histone H3K27me3, H3K4me3, and H3 ChIP-seq in TRB-ZF fusion T2 lines showing an early flowering phenotype as well as *fwa* as a control. Indeed, levels of H3K27me3 were increased, while H3K4me3 levels were reduced, in the TRB-ZFs over the *FWA* region

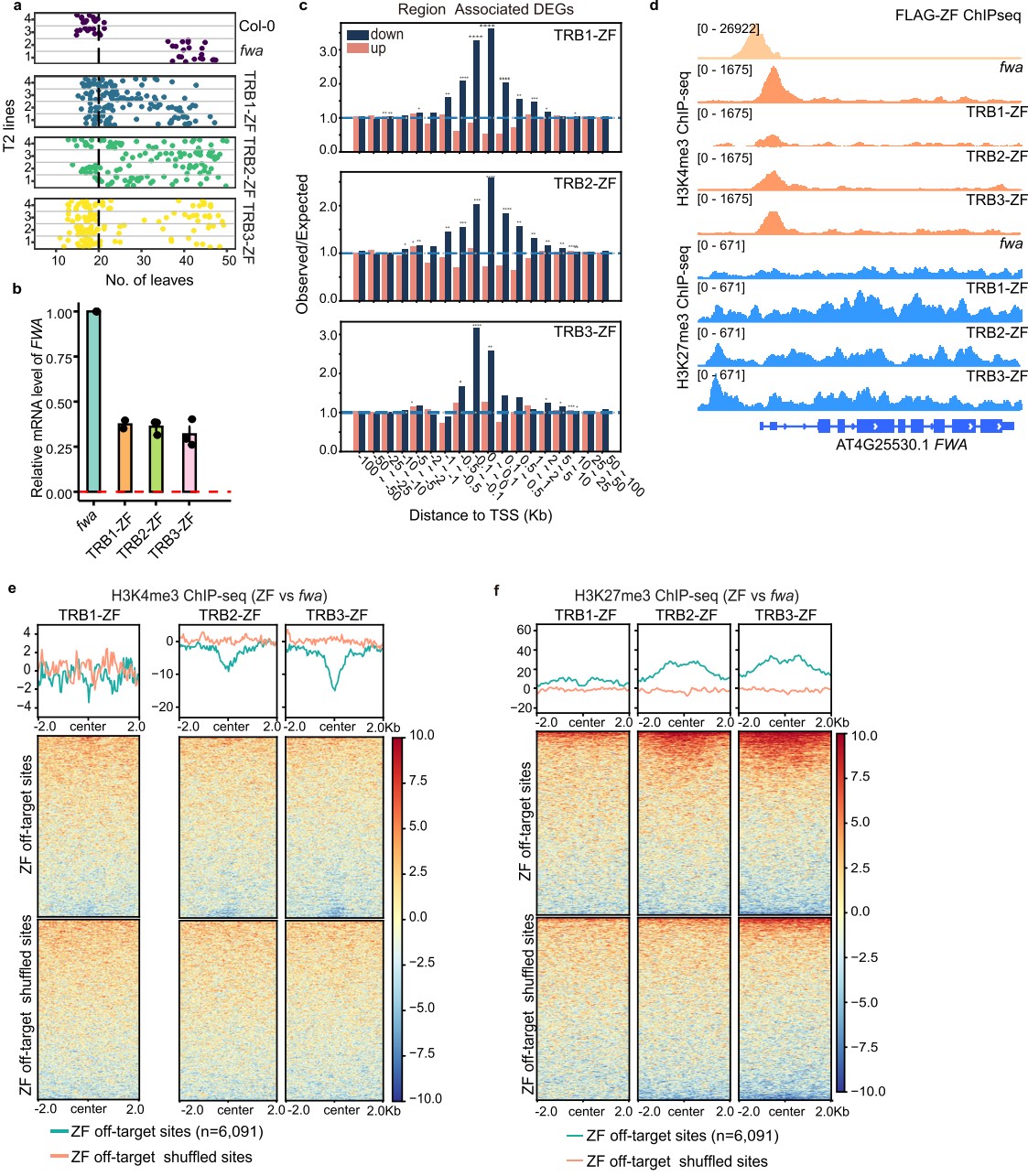

**Fig. 6 | Targeted gene silencing by TRB-ZF fusions with a combination of H3K27me3 deposition and H3K4me3 demethylation. a** Flowering time of *fwa*, Col-0, and four representative T2 lines of TRB1-ZF, TRB2-ZF, and TRB3-ZF. **b** Bar chart showing the relative mRNA level of *FWA* in *fwa* and TRB1, TRB2, and TRB3 T2 ZF fusion lines using normalized reads of RNA-seq data (RPKM). The error bars indicate the mean standard errors (SE) of the replicates (*n* = 3 biologically independent samples). **c** The observed/expected values of up- and down-regulated

DEGs in TRB-ZF T2 lines over ZF off-target sites (*n* = 6091), measured by RAD analysis. The asterisks indicate the *p* value calculated with hypergeometric test, \**p* < 0.05; \*\**p* < 0.01; \*\*\**p* < 0.001; and \*\*\*\**p* < 0.0001. **d** Screenshots of H3K4me3 and H3K27me3 ChIP-seq signals over the *FWA* region in *fwa*, TRB1-ZF, TRB2-ZF, and TRB3-ZF. The FLAG-ZF ChIP-seq signals indicated ZF binding site. **e-f** Metaplots and heatmaps showing normalized H3K4me3 and H3K27me3 ChIP-seq signals over ZF off target sites and shuffled sites (*n* = 6091) in *fwa*, TRB1-ZF, TRB2-ZF, and TRB3-ZF.

(Fig. 6d). This data suggests that changes in both histone modifications collectively silence *FWA* in TRB-ZFs. We also examined the ZF off target sites and found that for TRB2-ZF and TRB3-ZF, there was a large increase in H3K27me3 and a large decrease in H3K4me3, while for TRB1-ZF this effect was much smaller (Fig. 6e, f and Supplementary Fig. 14). Notably, there were some ZF off-target sites that failed to show gene silencing, H3K4me3 demethylation, or H3K27me3 deposition in the TRB-ZFs T2 transgenic lines, which might be due to pre-existing epigenetic features of these sites.

To determine whether the deposition of H3K27me3 or the removal of H3K4me3 were affected by pre-existing H3K27me3 or

H3K4me3, respectively, ZF off-target sites were divided into three clusters according to pre-existing levels of either H3K4me3 or H3K27me3 in *fwa* plants (Cluster 1 = high, Cluster 2 = medium, and Cluster 3 = low). Intriguingly, TRB-ZFs triggered H3K27me3 deposition mainly at H3K27me3 Cluster 1 of ZF off-target sites (Supplementary Fig. 15a, b), suggesting that TRB-ZFs preferred to ectopically deposit H3K27me3 at ZF off-target sites already somewhat enriched with H3K27me3. Furthermore, we found that TRB-ZFs triggered H3K4me3 removal mainly at regions with medium (H3K4me3 Cluster 2) or low (H3K4me3 Cluster 3) levels of pre-existing H3K4me3 (Supplementary Fig. 16a, b). To investigate whether this was due to selective

recruitment of JMJ14 by TRB-ZFs, Myc-JMJ14 was crossed with TRB-ZFs in order to perform Myc-JMJ14 ChIP-seq. As expected, a prominent Myc-JMJ14 peak appeared at *FWA* in TRB-ZFs, but not in the control (Supplementary Fig. 17a), confirming that TRBs recruit JMJ14 to *FWA*. Moreover, JMJ14 was mainly recruited to ZF off-target sites with medium or low levels of pre-existing H3K4me3 (Supplementary Fig. 17b, c), which is consistent with the result that TRB-ZFs triggered H3K4me3 removal mainly at H3K4me3 Cluster 2 and 3. Together with the data showing that TRB1 and JMJ14 co-localize to regions with low levels of H3K4me3 (Fig. 2b, c, and Supplementary Fig. 3c), these data suggest that TRBs can co-operate with JMJ14 to remove H3K4me2/3 at their co-bound sites, while TRBs fail to recruit JMJ14 to TRB1 unique binding sites enriched with H3K4me3. Collectively, these results demonstrate that TRB-ZFs can target H3K27me3 deposition and H3K4me3 removal at many sites in the genome.

Surprisingly, we found that H3K4me3 signals in TRB-ZFs were increased over ZF off-target sites with high levels of pre-existing H3K4me3 (Supplementary Fig. 16 a, b). However, this phenomenon was not correlated with H3K27me3, because H3K27me3 was not altered at ZF off-target sites with high levels of H3K4me3 (Supplementary Fig. 18a). A possible explanation is that TRBs are not efficient at recruiting JMJ14 to the H3K4me3 enriched regions, as the accessibility of JMJ14 might be inhibited by unknown mechanisms. Furthermore, our TRB IP-MS pulled down several peptides of WDR5A protein (Supplementary Data 1), which is one of the major components of the COMPASS-like complex that associates with H3K4 methyltransferases[42,43]. We analyzed WDR5A-ChIP-seq data from a recent paper[44], and found that WDR5A was mainly located at the H3K4me3 enriched regions (Supplementary Fig. 18b, c), and highly colocalized with TRB1 but not JMJ14 (Supplementary Fig. 18b, d). Collectively, these results suggest that TRB-ZFs can recruit JMJ14 to regions with low or medium levels of pre-existing H3K4me3 to remove methylation, and in addition, TRB-ZFs might also recruit WDR5A and associated H3K4 methyltransferases to regions with high levels of pre-existing H3K4me3 to add methylation. However, additional investigation is needed to confirm the dual regulation of H3K4me3 by TRBs at varied loci.

## Discussion

Previous work has shown that Arabidopsis TRBs mediate gene silencing by PRC2 mediated H3K27me3 deposition[5,8]. In this study, we showed that, in addition to the PRC2 complex, TRB proteins also recruit JMJ14 to demethylate the active chromatin mark H3K4me3. The TRBs thus serve as dual regulators of H3K27me3 deposition and H3K4me3 demethylation. We found that TRBs partially colocalized with JMJ14 over gene regions that show lower levels of H3K4me3, while both *trb1/2/3* and *jmj14-1* mutants exhibited an increased level of H3K4me3 over these regions. The increase of H3K4me3 triggered an up-regulation of target genes in both mutants, and a highly overlapping set of up-regulated genes was observed in *trb1/2/3* and *jmj14-1* mutants, consistent with TRBs and JMJ14 cooperating with each other to silence target genes at many sites.

It has been previously observed that H3K4me3 and H3K27me3 are in mutually exclusive domains of the Arabidopsis genome[13]; however, the underlying mechanism is not fully understood. Previous work in animals has shown that active histone marks such as H3K4me3 and H3K36me2/3 can inhibit the deposition of H3K27me3 to the same histone tail by PRC2[45,46]. In this study, we found that Arabidopsis TRB proteins form a complex with PRC2 and JMJ14 complexes, and the mutation of TRBs not only impacted H3K27me3 deposition, but also H3K4me3 demethylation, suggesting that TRBs might act upstream of PRC2 and JMJ14 complexes and partially contribute to the mutual exclusion of H3K4me3 and H3K27me3 at certain regions that are co-bound by TRB1 and JMJ14.

Using a gain of function approach, we found that tethering TRB proteins with an artificial zinc finger triggered gene silencing over the *FWA* region as well as many other ZF off-target regions. This silencing was associated with a gain of H3K27me3 and a decrease in H3K4me3, consistent with the other data suggesting that TRBs recruit both PRC2 and JMJ14 to synergistically suppress gene expression through H3K27me3 deposition and H3K4me3 demethylation at *FWA* and certain ZF off-target sites. However, we found that different ZF target sites displayed different behaviors depending on their pre-existing epigenetic state. Specifically, JMJ14 was mainly recruited to regions with lower levels of pre-existing H3K4me3 when ectopically expressed in TRB-ZFs backgrounds, which is consistent with endogenous TRBs and JMJ14 co-bound regions showing lower levels of H3K4me3, suggesting that TRBs prefer to recruit JMJ14 to H3K4me3 depleted regions. It will be interesting in the future to understand mechanistically why TRB is unable to stably recruit JMJ14 to regions that are already rich in H3K4me3.

## Methods

### Plant materials and growth conditions

The Arabidopsis plants used in this paper are *Arabidopsis thaliana* Col-0 ecotype, and the plants are grown under standard condition with 16 h light and 8 h dark, at 22 °C. The T-DNA insertion lines used in this study included *trb1-2* (SALK_001540), *trb2* (CS882628, SALKseq_4604), *trb3* (SALK_134645), *trb1/2/3* triple mutant, and *jmj14-1* (SALK_135712)[31]. The Agrobacterium (AGL0 strain) mediated floral dipping was used to generate all the transgenic plants.

### Plasmid construction

pTRB:TRB-FLAG, pJMJ14:JMJ14-FLAG, pJMJ14:JMJ14-Myc, and pTRB:TRB-FLAG-ZF108: the genomic DNA sequences of TRBs and JMJ14 with promoter sequences (around 2 kb upstream from the 5′UTR or until the next gene annotation) were first cloned into pENTR/D-TOPO vectors (Invitrogen), and then to the destination vector pEG302-GW-3XFLAG, pEG302-GW-9XMyc, or pEG302-GW-3XFLAG-ZF108 by LR reaction (LR Clonase II, Invitrogen). pUBQ10:ZF108-FLAG-TRB3: the cDNA sequence of TRB3 was cloned into pENTR/D-TOPO vectors (Invitrogen), and then to the destination vector pMDC123-UBQ10:ZF-3XFLAG-GW by LR reaction (LR Clonase II, Invitrogen).

### Native immunoprecipitation and mass spectrometry

Around ten grams of floral buds from two biological replicates of Col-0 control and FLAG-TRB1, 2, and 3 were ground into fine powder with liquid nitrogen and mixed with 25 mL IP buffer (50 mM Tris-HCl pH 8.0, 150 mM NaCl, 5 mM EDTA, 10% glycerol, 0.1% Tergitol, 0.5 mM DTT, 10 mM MG-132 (Sigma), 1 mM PMSF, 1 μg/ml pepstatin, and 1x Protease Inhibitor Cocktail (Roche)). The lysate was rotated at 4 °C for 10 minutes and the tissue was disrupted with dounce homogenizer until lump-free. The lysate was then filtered through double layers of Miracloth and centrifuged at 20,000×g for 10 min at 4 °C. The supernatant was incubated with 250 μL pre-washed (by IP buffer) and pre-blocked (by 5% BSA for 15 min) anti-FLAG M2 magnetic beads (Sigma) at 4 °C for 2 h. The magnetic beads were washed 4 times with IP buffer and eluted with TBS buffer containing 250 μg/mL 3X FLAG peptides (sigma). The elusion was precipitated with trichloroacetic acid (Sigma) and subject to Quantitative proteomics (MS−MS).

### Quantitative proteomics

Protein pellets were resuspended in 8 M urea and 100 mM Tris pH 8.5, then reduced by adding Tris (2-carboxyethyl) phosphine (TCEP) to a final concentration of 5 mM and incubation for 30 min. Next, the proteins were alkylated by adding iodoacetamide to a final concentration of 10 mM for another 30 min at room temperature. Before protein digestion, the urea concentration was diluted to 2 M with 100 mM Tris pH 8.5. Then, the proteins were digested with LysC (BioLabs) at a 1:100 enzyme/protein ratio at 37 °C for 4 hours, followed by the trypsin digestion at 1:100 (trypsin: protein) at 37 °C for 12 h. To stop the

digestion, 5% formic acid was added to the samples. Next, the peptides were desalted using C18 pipette tips (Thermo Scientific) and reconstituted in 5% formic acid before being analyzed by LC-MS/MS. Tryptic peptide mixtures were loaded onto a 25 cm long, 75 μm inner diameter fused-silica capillary, packed in-house with bulk 1.9 μM ReproSil-Pur beads with 120 Å pores as described[47]. The peptides were delivered by a 140-min water-acetonitrile linear gradient in 6–28% buffer (acetonitrile solution, 0.1% formic acid and 3% DMSO) using a Dionex Ultimate 3000 nanoflow UHPLC (Thermo Scientific), at a flow rate of 200 nL/min, further increased to 35% and followed by a rapid ramp-up to 85%. The eluted peptides were ionized and the Orbitrap Fusion Lumos Tribrid Mass Spectrometer (Thermo Scientific) was used to acquire the Mass Spectrometer. The data-dependent acquisition strategy consisted of a repeating cycle of a full MS1 spectrum (Resolution = 120,000) followed by sequential MS2 scan (Resolution = 15,000).

Label-free quantification was performed using the MaxQuant software package (v1.6.17.0) with LFQ default setting[48], and Arabidopsis TAIR 10 proteome database was used for the database search. Trypsin digestion was applied and a maximum of two missed cleavages were allowed in all searches for tryptic peptides of length 8–40 amino acids. In all, 1% false discovery rate was used as a filter at both protein and peptide-spectrum match (PSM) levels. The mass spectrometry proteomics data have been deposited to the ProteomeXchange Consortium via the MassIVE partner repository, and the accession number is MSV000091349.

## Cross-linking IP-MS

Ten grams of Arabidopsis floral bud tissues from one biological replicate of Col-0 control and FLAG-TRBs TRB1, 2, and 3 were ground into fine powder with liquid nitrogen and resuspended with nuclei isolation buffer (50 mM HEPES, 1 M sucrose, 5 mM KCl, 5 mM MgCl$_2$, 0.6% Triton X-100, 0.4 mM PMSF, 5 mM benzamidine, 1% formaldehyde (Sigma), and 1x Protease Inhibitor Cocktail (Roche)). The lysate was rotated at room temperature for 10 minutes and the cross-linking was terminated by adding 1.7 mL 2 M fresh-prepared glycine solution. Then the lysate was filtered through one layer of Miracloth and centrifuged at 4 °C with 1500×g for 10 min. The nuclei were then washed twice with NRBT buffer (20 mM Tris-HCl pH 7.5, 2.5 mM MgCl$_2$, 25% glycerol, and 0.2% Triton X-100), and resuspended with 6 ml of RIPA buffer (1x PBS, 1% NP-40, 0.5% sodium deoxycholate, and 0.1% SDS). Next, the resuspended solution was subjected to sonication for 22 cycles (30 s on/30 s off per cycle) with Bioruptor Plus (Diagenode). The lysate was centrifuged with 8000×g at 4 °C for 15 min, and supernatant was incubated with 250 μL pre-washed and blocked (by 5% BSA for 15 minutes) anti-FLAG M2 magnetic beads (Sigma) at 4 °C for 2 h. The subsequent experiment procedures are the same as native IP-MS above as described above.

## Co-immunoprecipitation

One gram of 2-week-old seedling was collected from FLAG-TRB1xMyc-JMJ14, FLAG-TRB2xMyc-JMJ14, FLAG-TRB3xMyc-JMJ14, FLAG-JMJ14x Myc-TRB1, Myc-JMJ14, and Myc-TRB1. The seedlings were ground into fine powder with liquid nitrogen and mixed with 10 mL IP buffer (50 mM Tris HCl pH 7.5, 150 mM NaCl, 2 mM EDTA, 2 mm DTT, 1% TritonX-100, and 1x Protease inhibitor (Roche)), and incubated at 4 °C for 20 min. The lysate was centrifuged at 18,000×g for 20 min at 4 °C, and the supernatant was filtered through a double layer Miracloth (repeat the centrifuging and filtering once). The supernatant was incubated with 30 μL anti-FLAG M2 Affinity Gel (Sigma) for 2 h at 4 °C. The beads were washed with IP buffer for 5 times, and proteins were eluted with 40 μL elution buffer (IP buffer containing 100 μg/mL 3xFLAG peptide as final concentration) by vigorously shaking at 4 °C for an hour. The beads were then centrifuged at 1,600x g at 4 °C for 5 min and the supernatant was mixed with 5x SDS loading buffer for western blot.

## Bimolecular fluorescence complementation

The N-terminal fragment of YFP was fused with JMJ14 and NAC052, and the C-terminal fragment of YFP was fused with TRBs. These plasmids were transformed into Agrobacteria strain AGL0 for the transient co-expression assay in 3-4 weeks old *Nicotiana benthamiana* plants. The YFP signals of the infiltrated leaves of *Nicotiana benthamiana* were examined using confocal microscopy (Zeiss).

## Chromatin immunoprecipitation sequencing

Two grams of 4–5-week-old leaves were used for TRB-ZFs histone ChIPs; two grams of leaf tissue of 3-4-week-old Myc-JMJ14 transgenic lines in Col-0 and TRB-ZFs backgrounds were used for Myc ChIPs; two grams of flower buds were used for FLAG-TRB and JMJ14 ChIPs; and one-gram seedlings of *jmj14-1* and *trb1/2/3* mutants were used for histone ChIPs. The plant materials were ground into fine powder with liquid nitrogen and resuspended with nuclei isolation buffer (50 mM HEPES, 1 M sucrose, 5 mM KCl, 5 mM MgCl2, 0.6% Triton X-100, 0.4 mM PMSF, 5 mM benzamidine, 1% formaldehyde (Sigma), and 1X Protease Inhibitor Cocktail (Roche)) for 10 min. 1.7 ml of 2 M fresh-made glycine solution was added to terminate the crosslinking reaction. The lysates were filtered through one layer of Miracloth and the nuclei were collected by centrifuge at 4 °C with 2880×g for 20 min. The nuclei were resuspended with extraction buffer 2 (0.25 M sucrose, 10 mM Tris-HCl pH 8.0, 10 mM MgCl$_2$, 1% Triton X-100, 5 mM BME, 0.1 mM PMSF, 5 mM Benzamidine, and 1x Protease Inhibitor Cocktail (Roche)), centrifuge with 12,000 x g at 4 °C for 10 minutes, and then resuspended with extraction buffer 3 (1.7 M sucrose, 10 mM Tris-HCl pH 8.0, 2 mM MgCl$_2$, 0.15% Triton X-100, 5 mM BME, 0.1 mM PMSF, 5 mM Benzamidine, and 1x Protease Inhibitor Cocktail (Roche)), at 4 °C with 12,000×g for 60 min. The nuclei were lysed with nucleic lysis buffer (50 mM Tris-HCl pH 8.0, 10 mM EDTA, 1% SDS, 0.1 mM PMSF, 5 mM Benzamidine, and 1x Protease Inhibitor Cocktail (Roche)) on ice for 10 minutes and the lysate was sheared by Bioruptor Plus (Diagenode) for 22 cycles (30 s on/30 s off per cycle). The lysate was centrifuged twice at 4 °C with 20,000×g for 10 min, and the supernatant was incubated with antibody at 4 °C overnight. Next, the magnetic Protein A and Protein G Dynabeads (Invitrogen) were added and the mixture was rotated at 4 °C for 2 h. The beads were washed with low salt solution twice (150 mM NaCl, 0.2% SDS, 0.5% Triton x-100, 2 mM EDTA, and Tris pH 8.0), high salt solution (150 mM NaCl, 0.2% SDS, 0.5% Triton x-100, 2 mM EDTA, and Tris pH 8.0), LiCl solution (250 mM LiCl, 1% Igepal, 1% Sodium Deoxycholate, 1 mM EDTA, and 10 mM Tris pH 8.0), and TE solution (1 mM EDTA and 10 mM Tris pH 8.0) for 5 minutes at 4 °C, respectively. The chromatin was eluted with elution buffer (1% SDS, 10 mM EDTA, and 0.1 M NAHCO$_3$), and subjected to reverse crosslinking at 65 °C overnight. Then 1 μL Protease K (20 mg/mL, Invitrogen), 10 μL of 0.5 M EDTA pH 8.0, and 20 μL 1 M Tris pH 6.5 were added to deactivate the protein (45 °C, 4 h) and the DNA was purified through phase lock gel (VWR) and precipitated with 1/10 volume of 3 M Sodium Acetate (Invitrogen), 2 μL GlycoBlue (Invitrogen), and 1 mL 100% Ethanol at −20 °C overnight. The precipitated DNA was used for library construction following the manual of Ovation Ultra Low System V2 kit (NuGEN), and the libraries were sequenced on an Illumina NovaSeq 6000 sequencer.

## RNA-seq

Two-week-old seedlings of Col-0 wild type, *trb1/2/3*, and *jmj14-1 mutants*, and four-week-old leaves of *fwa* and TRB-ZF T2 transgenic lines with early flowering phenotype, were used for RNA extraction using Direct-zol RNA MiniPrep kit (Zymo). The RNA-seq libraries were constructed with 1 μg of RNA per sample by following the manual of TruSeq Stranded mRNA kit (Illumina). The RNA-seq libraries were sequenced on an Illumina NovaSeq 6000 or an Illumina HiSeq 4000 sequencer.

## Whole genome bisulfite sequencing

Two-week-old Arabidopsis seedlings of Col-0 wild type and *trb1/2/3* triple mutants were used for DNA extraction using DNeasy Plant Mini Kit (QIAGEN). A total of 500 ng DNA was sheared with Covaris S2 (Covaris) into around 200 bp at 4 °C. The DNA fragments were used to perform end repair reaction using the Kapa Hyper Prep kit (Roche), and together with TruSeq DNA Sgl Index Set A/B (Illumina) to perform adapter ligation. The ligation products were purified with AMPure beads (Beckman Coulter), and then converted with EpiTect Bisulfite kit (QIAGEN). The converted ligation products were used as templates, together with the primers from the Kapa Hyper Prep kit (Roche) and MyTaq Master mix (Bioline), to perform PCR. The PCR products were purified with AMPure beads (Beckman Coulter) and sequenced by an Illumina NovaSeq 6000 sequencer.

## BS-PCR-seq

The leaves of four to five-week-old Col-0 wild type, *fwa*, and the T2 transgenic lines of TRB-ZF fusions showing early flowering phenotype were used to perform bisulfite PCR at *FWA* promoter. Genomic DNA was extracted with DNeasy Plant Mini Kit (QIAGEN) and converted with EpiTect Bisulfite kit (QIAGEN), and then was used as a template to amplify three regions over *FWA* promoter, including Region 1 (chr4: 13038143-13038272), Region 2 (chr4: 13038356- 13038499), and Region3 (chr4: 13038568-13038695). Pfu Turbo Cx (Agilent), dNTP (Takara Bio), and the primers designed for the above-mentioned *FWA* regions (see Supplementary Data 5) were used for PCR. The PCR products were purified with AMPure beads (Beckman Coulter), and then used to construct libraries with Kapa HyperPrep Kit (Roche). The libraries were sequenced on an Illumina iSeq 100 sequencer.

## Bioinformatic analysis

ChIP-seq analysis: Trim_galore (https://www.bioinformatics.babraham.ac.uk/projects/trim_galore/) was used to trim the ChIP-seq raw reads. The trimmed reads were aligned to TAIR10 genome using Bowtie 2 version 2.3.4.3[49], which allowed one unique mapping site with 0 mismatch. The Samtools version 1.9 was used to remove the duplicated reads[50], and the deeptools version 3.1.3 was used to generate tracks[51], which was normalized by using RPKM. MACS2 version 2.1.1. was used for peak calling, and the hyperchipable sites were filtered. The hyperchipable sites were the peaks appeared in multiple ChIP-seq replicates of Col-0.

RNA-seq analysis: The bowtie2 version 2.3.4.3[49] was used to align the raw reads of RNA-seq data to the TAIR10 transcriptome and rsem-calculate-expression of RSEM was used to calculate the expression levels with default settings[52]. The Samtools version 1.9[50] was used to generate tracks and the bamCoverage of deeptools version 3.1.3[51] was used to normalize the data with RPKM. The Trinity version 2.8.5[53] was used to call DEGs. The cut-off of DEGs was set as log2 FC ≥ 1 and FDR < 0.05.

WGBS analysis: the WGBS raw reads were aligned to both strands of reference genome TAIR10 using BSMAP (v.2.74)[54], and the alignment allowed up to 2 mismatches and 1 best hit. The reads with more than 3 consecutives methylated CHH sites were removed, and the methylation level was calculated with the ratio of C/(C + T). Figure 5a showed the methylation levels at 1 kb flanking regions of RdDM target sites (Pol V peaks)[55] in Col-0 wild type and *trb1/2/3* triple mutant.

BS-PCR-seq analysis: The raw reads were aligned to TAIR10 using BSMAP (v.2.74)[54], which allowed up to two mismatches and one best hit. The cytosines that were covered less than 20 times, as well as the reads that had more than three consecutives methylated CHH sites, were removed. The ratio of C/(C + T) was calculated as the methylation level of each cytosine within the designed *FWA* regions, and the plots were made by using customized R scripts.

Binding motif analysis: findMotifsGenome.pl from HOMER was used to discover the motifs of the ChIP-seq data sets[56]. Homer2 scanMotifGenomeWide.pl was used to scan genome-wide distributions of motifs TAAGGG and CTTGNNNNNCAAG, and then the bedtools intersect was used to identify the TRB1 and JMJ14 co-bound peaks and shuffled peaks containing TAAGGG and/or CTTGNNNNNCAAG motifs[57].

## Reporting summary

Further information on research design is available in the Nature Portfolio Reporting Summary linked to this article.

## Data availability

Source data of Figs.1a and 6c, and Supplementary Figs. 1a and 11b are provided with this paper. All the high-throughput sequencing data generated in this study are accessible at NCBI's Gene Expression Omnibus (GEO) via GEO Series accession number GSE204681. The mass spectrometry proteomics data have been deposited to the ProteomeXchange Consortium via the MassIVE partner repository, and the accession number is MSV000091349. Source data are provided with this paper.

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

## Acknowledgements

We thank Dr. Franziska Turck for information on their previously published *trb1/2/3* triple mutant allele; Dr. Colette Picard for advice on *trb1/2/3* mutant imaging; Dr. Ranjith Papareddy for comments of the manuscript; and Peggy Hsuanyu Kuo, Tiffany Dong, and Soo-Young Park for technical support. We also thank Mahnaz Akhavan and the UCLA BSCRC BioSequencing Core for sequencing support. This work was supported by a Bill and Melinda Gates Foundation grant (OPP1125410) to S.E.J.; S.E.J. is a Howard Hughes Medical Institute Investigator.

## Author contributions

M.W. and S.E.J. designed the research, interpret the data, and wrote the manuscript; M.W. performed most of the experiments; J.G.B. generated TRB2 and TRB3 ZF fusion transgenic lines; Z.Z. and M.W. performed bioinformatic data analysis; M.W., Y.J.A., and J.W. performed IP-MS and interpreted the data. Y.S. and K.W. performed BiFC assay. J.C.R., M.L., C.N., and J.Z. provided technical help. M.W. and S.F. performed BS-PCR-seq and high throughput sequencing.

## Competing interests

The authors declare no competing interests.
