## [Peer Review File · Nature Communications]

Arabidopsis TRB proteins function in H3K4me3 demethylation
by recruiting JMJ14.Reviewer #1 (Remarks to the Author):

The manuscript entitled "Arabidopsis TRB proteins function in H3K4me3 demethylation by recruiting JMJ14" describes the role of TRB proteins, which are known PRC2-recruiting transcription factors, in recruiting JMJ14, an H3K4me3 demethylase, and thus in coordinating H3K27me3 deposition with H3K4me3 removal. In the work, the authors found two groups of TRB targets. The first group named Cluster 1 harbors common targets between TRBs and JMJ14, whereas the second group named Cluster 2 is composed of TRB but not JMJ14 targets. Cluster 1 (n=892), which is much smaller than Cluster 2 (n=6,710), showed a nice correlation between the role of TRB and H3K4me3 decrease. However, in the larger group, Cluster 2, the role of TRB was not well correlated with H3K4me3 level (Figure 3), indicating JMJ14-independent H3K4me3 demethylase activity at these loci.

From the discovery of interaction between TRBs and JMJ14 to the demonstration of these two factors in regulating epigenetic status and expression of Cluster 1 genes, is very clear and persuasive. This, however, falls short to explain genome-wide anti-correlation between H3K27me3 and H3K4me3, which is also observed in Cluster 2 loci. So TRB-mediated recruitment of JMJ14 seems to explain a small part of the anti-correlation between H3K27me3 and H3K4me3 observed in the Arabidopsis epigenome. However, current abstract and major conclusions are written too broadly and do not provide information on the aspects unexplainable by this work.

PRC2 is known to be recruited by VAL1/2 and BPC transcription factors as well as by TRBs. Based on recent publication (NAR 49:98, 2021), VALs might have a broader effect than TRBs in PRC2 recruitment. In addition, JMJ14 forms a clade of Jumonji-family H3K4me demethylases with five other members. These facts, which are not mentioned in this manuscript, have potentials to be related to the unexplainable aspects of the work. Efforts to test the roles of other PRC2 recruiters and JMJ members may lead to more general and broad conclusions.

I do not know if the final part of the work in which the authors used an artificial ZnF-dependent targeting of TRBs is meaningful. The targeting was originally designed for the FWA locus. However, the authors explain results obtained from broad non-specific off-target effects. If these would gain a meaning, the authors should re-demonstrate the effects observed through specific targeting.

Below are a few minor points:

1. Line 18: ...H3K27me3 and H3K4me3 are mutual exclusive withing plant genomes... has to be ...H3K27me3 and H3K4me3 are mutually exclusive within plant genomes...
2. Line 62: H3K27eme3 has to be H3K27me3
3. Line 80: Note that the first report on JMJ14 as H3K4 demethylase was PLoS ONE 4:e8033, 2009.
4. Add titles for Supplementary Tables.
5. Supplementary Fig. S1b: There is an error in figure label.
6. Supplementary Fig. S3: Please provide E-value for each predicted motif.
7. Supplementary Fig. S7c: Error in figure label.

Reviewer #2 (Remarks to the Author):

In this manuscript, Wang et al identify a new role of the Arabidopsis Telomere Repeat Binding proteins TRB1, TRB2 and TRB3 in the regulation of histone modification by triggering sequence-specific recruitment of the JMJ14 demethylase. Building on previous knowledge that TRBs can recruit PRC2 to telomeric motifs, the authors propose a model in which these proteins can, on one hand, trigger the removal of the active H3K4me3 chromatin mark, and on the other hand trigger the establishment of a repressive chromatin status through H3K27me3 deposition. That dual function is nicely supported by artificially targeting TRB1 to new loci upon protein engineering with a sequence-specific Zinc-Finger domain. The manuscript is very well written, nicely illustrated and should be of great interest for the readership of Nature Communications.

As detailed below, at several places it falls short in description of interpretation of the findings.

- I sincerely regret that no token was provided to the reviewers for a fair assessment of the NGS data quality

- The notion that a single transcription factor like TRB1 can drive H3K4me3 erasure and H3K27me3 deposition is nicely defined in the general context of these two chromatin marks being generally mutually exclusive in the Arabidopsis epigenome. Yet, the proposed mechanism can only explain a small portion of this observation, as TRBs do not define the majority of PRC2 target genes. This aspect should be cautiously described. Ideally, it would be good to support the statement by an assessment of the proportion of loci at which TRB-JMJ14 association could trigger H3K4me3-to-H3K27me3 switches.

- TRB1, 2 and 3 protein-protein association with JMJ14 is only supported by co-IP. This assay should be complemented by other approaches to determine whether interactions are direct (e.g. Y2H). They should also be confirmed to occur in planta (e.g., subnuclear co-localization), as co-IP positive results can result from biased associations occurring in cell extracts. Using a crosslinker does not necessarily help avoiding this.

- As shown in Figure S1b, JMJ14 activity with TRB1, 2, 3 is supported by the partial co-distribution of their chromatin association profiles along the genome. Yet, this analysis does not inform on whether the chromatin associations occur on the same loci in the same cells. This analysis could first be completed by stating the proportion of JMJ14 peaks that is linked to a TRB-associated gene and, vice versa, to which extent TRB1, 2, 3 peaks correspond to JMJ14-associated genes. More importantly, given the observation that TRBs are often bound to chromatin independently of JMJ14, the latter correlation analysis should be completed by an experimental assessment of JMJ14 chromatin profile in *trb1,2,3* loss-of-function plants. This is a difficult but realistic experiment that can be achieved upon introgressing the FLAG-JMJ14 transgene in the available *trb* triple mutant plants in which one of the *trb* allele is in a heterozygous state. This would allow testing properly the proposed model by determining whether JMJ14 actually relies on TRBs for its recruitment at specific loci. Alternatively, the authors may complement the TRB1-ZF strategy by determining whether ectopic targeting of TRB1 to FWA and to all other off-targets results in systematic recruitment of JMJ14 to these loci.

- TRBs have been shown to display the intrinsic property to associate to Telobox motifs, while JMJ14 associates with NAC050 and NAC052 that typically bind CTTGnnnnnCAAG motifs. The conclusion of TRBs and JMJ14 acting together theoretically implies that loci at which the TRB-JMJ14 mechanism operates display both a Telobox motif and a NAC050/2 binding motif. Is-this observed? If many exceptions are found, does-it mean that TRB-mediated JMJ14 recruitment to Telobox-containing genes acts independently of NAC050/2 proteins at loci bearing a Telobox but not a CTTGnnnnnCAAG motif? Investigating this aspect could allow addressing a central aspect of the findings, as to whether JMJ14 is recruited by TRBs to favor H3K27me3 as a main function, or whether JMJ14 has other functions independently of TRBs relying on NAC050/2 but no TRB protein association.

- A follow-up question is whether JMJ14 can associate with TRBs and to NAC050/2 in the same protein complex?

- The model predicts that ectopic targeting of TRB1-ZF to FWA or to any other loci will locally remove/reduce H3K4me3 and increase H3K7me3. This is what is shown in Figures 6, S10, S11 for a few loci where the model nicely works. Based on the authors' observations, is-it a general observation, and how does this compare when ZF off-targets do no, bear H3K4me3 and/or already bear H3K27me3 in *fwa* reference plants?

- To support the claim of the new *trb1/2/3* mutant line having different phenotypes than previously published lines, the development of the different types of mutant lines should be compared upon growth in the same conditions.

- The rationale to conduct DNA methylation analyses in the *trb1/2/3* mutant line is obscure to me. How does-this relate to H3K4me3 removal and JMJ14 function at transcribing genes?

- The observation of CHH and CHG methylation loss in the *trb1/2/3* mutant line prompts the question as to whether TRB1-ZF expression in wild-type plants, which bear methylated FWA, can decrease the DNA methylation state of the FWA locus and influence flowering time.

-Last, the data indicate that TRB1/2/3 can also associate to domains stably enriched in H3K4me3. At these loci, TRBs most likely do not operate in H3K4me3 demethylation but may rather trigger H3K27me3 removal. This hypothesis is supported by the authors' MS data in which REF6 is one of the most abundant proteins co-purifying with TRB1, 2 and 3. Can the authors comment on this and potentially elaborate more in the conclusions?

Minor points

- Lane 52, Arabidopsis has more than three TRB proteins, but only 3 have a clear coiled-coil domain.

- Lane 54, TRB Myb-like domain has also been shown previously to enable TRB binding to telomeric motifs outside of telomeres

- Lane 62, "TRB1, 2, 3 are close homologs" is ambiguous. Is-it meant that they share a high sequence similarity or just that they functionally overlap?

- Lane 82, why mention TRB recruitment of PRC1/2 complexes? Is it meant PRC2 and/or LHP1?

- Figure S4 should compare *trb1/2/3* mutant phenotypes to WT plants, especially because the genetics suggests that TRB1, 2 and 3 genes may have a dosage effect in *trb1/2/3+/-* plants.

- To support the claims at lane 154, figure S4 should provide information on the proper genotypes used, and their confirmation for each of the spotted individuals.

We would like to thank both reviewers for the very helpful and constructive comments on our manuscript. **The reviewer comments are in bold type** and our response is in regular type. We hope the manuscript is now acceptable for publication in *Nature Communications*.

REVIEWER COMMENTS

Reviewer #1 (Remarks to the Author):

The manuscript entitled “Arabidopsis TRB proteins function in H3K4me3 demethylation by recruiting JMJ14” describes the role of TRB proteins, which are known PRC2-recruiting transcription factors, in recruiting JMJ14, an H3K4me3 demethylase, and thus in coordinating H3K27me3 deposition with H3K4me3 removal. In the work, the authors found two groups of TRB targets. The first group named Cluster 1 harbors common targets between TRBs and JMJ14, whereas the second group named Cluster 2 is composed of TRB but not JMJ14 targets. Cluster 1 (n=892), which is much smaller than Cluster 2 (n=6,710), showed a nice correlation between the role of TRB and H3K4me3 decrease. However, in the larger group, Cluster 2, the role of TRB was not well correlated with H3K4me3 level (Figure 3), indicating JMJ14-independent H3K4me3 demethylase activity at these loci.

Form the discovery of interaction between TRBs and JMJ14 to the demonstration of these two factors in regulating epigenetic status and expression of Cluster 1 genes, is very clear and persuasive. This, however, falls short to explain genome-wide anti-correlation between H3K27me3 and H3K4me3, which is also observed in Cluster 2 loci. So TRB-mediated recruitment of JMJ14 seems to explain a small part of the anti-correlation between H3K27me3 and H3K4me3 observed in the Arabidopsis epigenome. However, current abstract and major conclusions are written too broadly and do not provide information on the aspects unexplainable by this work.

Thank you for this comment. We agree with the reviewer that TRB-JMJ14 connection can only explain a small portion of the mutual exclusion between H3K4me3 and H3K27me3, and we have deleted all the broad claims in the revision as suggested.

We have added “We demonstrate that TRB proteins not only recruit PRC2 complexes to deposit H3K27me3^{1,2}, but also recruit JMJ14 to remove H3K4me3, which can partially explain the mutual antagonism of these two histone marks over certain regions that are co-bound by JMJ14 and TRBs.” (Page 2, line 84-87) in the introduction.

We also have added, “It has been previously observed that H3K4me3 and H3K27me3 are in mutually exclusive domains of the Arabidopsis genome³; however, the underlying mechanism is not fully understood. Previous work in animals has shown that active histone marks such as H3K4me3 and H3K36me2/3 can inhibit the deposition of H3K27me3 to the same histone tail by PRC2^{4,5}. In this study, we found that Arabidopsis TRB proteins form a complex with PRC2 and JMJ14 complexes, and the mutation of TRBs not only impacted H3K27me3 deposition, but also H3K4me3 demethylation, suggesting that TRBs might act upstream of PRC2 and JMJ14 complexes and partially contribute to the mutual exclusion of H3K4me3 and H3K27me3 at certain regions that are co-bound by TRB1 and JMJ14.” (Page 8, line 332-340) in the discussion.

In addition, we have added experiments to further address this point. We crossed Myc-JMJ14 to TRB-ZF in order to perform Myc ChIP-seq, which showed that TRB-ZFs could recruit JMJ14 to the *FWA* locus and to many ZF off-target sites with medium or low levels of pre-existing H3K4me3 (Supplementary Fig. 17). This is consistent with our observation that TRB-ZFs triggered H3K4me3 removal mainly at ZF off target sites with medium or low levels of pre-existing H3K4me3 (Supplementary Fig. 16). Together with TRB1 and JMJ14 co-bound regions showing low levels of H3K4me3 (Fig. 2b-c, and Supplementary Fig. 3c), these data suggest that TRBs can co-operate with JMJ14 for removal of H3K4me2/3 at their co-bound loci, while TRB failed to recruit JMJ14 to the TRB1 unique binding sites enriched with H3K4me3.

We have added the following associated text “To determine whether the deposition of H3K27me3 or the removal of H3K4me3 were affected by pre-existing H3K27me3 or H3K4me3, respectively, ZF off-target sites were divided into three clusters according to pre-existing levels of either H3K4me3 or H3K27me3 in *fwa* plants (Cluster 1 = high, Cluster 2 = medium, and Cluster 3 = low). Intriguingly, TRB-ZFs triggered H3K27me3 deposition mainly at H3K27me3 Cluster 1 of ZF off-target sites (Supplementary Fig. 15a, b), suggesting that TRB-ZFs preferred to ectopically deposit H3K27me3 at ZF off-target sites already somewhat enriched with H3K27me3. Furthermore, we found that TRB-ZFs triggered H3K4me3 removal mainly at regions with medium (H3K4me3 Cluster 2) or low (H3K4me3 Cluster 3) levels of pre-existing H3K4me3 (Supplementary Fig. 16a, b). To investigate whether this was due to selective recruitment of JMJ14 by TRB-ZFs, Myc-JMJ14 was crossed with TRB-ZFs in order to perform Myc-JMJ14 ChIP-seq. As expected, a prominent Myc-JMJ14 peak appeared at *FWA* in TRB-ZFs, but not in the control (Supplementary Fig. 17a), confirming that TRBs recruit JMJ14 to *FWA*. Moreover, JMJ14 was mainly recruited to ZF off-target sites with medium or low levels of pre-existing H3K4me3 (Supplementary Fig. 17b, c), which is consistent with the result that TRB-ZFs triggered H3K4me3 removal mainly at H3K4me3 Cluster 2 and 3. Together with the data showing that TRB1 and JMJ14 co-localize to regions with low levels of H3K4me3 (Fig. 2b, c, and Supplementary Fig. 3c), these data suggest that TRBs can co-operate with JMJ14 to remove H3K4me2/3 at their co-bound sites, while TRBs fail to recruit JMJ14 to TRB1 unique binding sites enriched with H3K4me3.” (Page 7, line 284-302).

PRC2 is known to be recruited by VAL1/2 and BPC transcription factors as well as by TRBs. Based on recent publication (NAR 49:98, 2021), VALs might have a broader effect than TRBs in PRC2 recruitment. In addition, JMJ14 forms a clade of Jumonji-family H3K4me demethylases with five other members. These facts, which are not mentioned in this manuscript, have potentials to be related to the unexplainable aspects of the work. Efforts to test the roles of other PRC2 recruiters and JMJ members may lead to more general and broad conclusions.

Thank you for this excellent point, we have added the information about VAL1/2, BPC, AZF1, and the other H3K4 demethylases (JMJ14-18, LDL1-3 and FLD) into the introduction.

We have added “In plants, the PRC2 complex can be recruited by VIVIPAROUS1/ABI3-LIKE1/2 (VAL1/2), BASIC PENTACYSTEINE 1 (BPC1), and ARABIDOPSIS ZINC FINGER1 (AZF1), in addition to TRBs⁶⁻⁸. The removal of H3K4me3 is controlled by H3K4 demethylases, which include Jumonji-domain containing proteins (JMJ14-18)⁹⁻¹³, and Lysine-Specific Demethylase 1 Like proteins, LDL1-3 and FLD (FLOWERING LOCUS D)¹⁴⁻¹⁶. JMJ14 is a well-studied H3K4 demethylase that removes H3K4 di- and tri-methylation and forms a complex with two NAC type transcription factors, NAC050 and NAC052^{9, 17-19}.” (Page 2, line 75-81).

We also agree with the reviewer that testing the role of the other PRC2 recruiters and JMJ members may lead to more general and broader conclusions. However, our TRB IP-MS only pulled down JMJ14 but not the other H3K4 demethylases, and our manuscript is mainly focused on the function of TRBs and JMJ14. Therefore, as mentioned above, we have deleted our general and broad conclusions concerning the antagonistic relationship between H3K4me3 and H3K27me3. We hope this will address the reviewer's concern.

I do not know if the final part of the work in which the authors used an artificial ZnF-dependent targeting of TRBs is meaningful. The targeting was originally designed for the FWA locus. However, the authors explain results obtained from broad non-specific off-target effects. If these would gain a meaning, the authors should re-demonstrate the effects observed through specific targeting.

We agree that as a biotechnology tool, ZFs have the disadvantage of binding to many locations in the genome, relative to CRISPR/Cas9 which binds very specifically. Thus, we have removed the sentences that suggest the application of TRB-ZF as a tool in epigenome engineering. However, as a research tool, and for this study in particular, zinc fingers are in some ways superior. The fact ZF108 could be used at *FWA* to study silencing, but also used at many other loci to confirm the silencing effect is a big advantage of using zinc fingers over CRISPR or other systems that can only look at one locus at a time. Therefore, we feel that the use of ZF for targeting in this study makes the results more meaningful because it validates the result at many loci.

Below are a few minor points:

1. Line 18: ...H3K27me3 and H3K4me3 are mutual exclusive withing plant genomes... has to be ...H3K27me3 and H3K4me3 are mutually exclusive within plant genomes...

Thanks for this comment. We have deleted this sentence in the abstract to avoid broad claims, and we have changed mutual exclusive to mutually exclusive throughout the manuscript.

2. Line 62: H3K27eme3 has to be H3K27me3

Done, thanks.

3. Line 80: Note that the first report on JMJ14 as H3K4 demethylase was PLoS ONE 4:e8033, 2009.

Thank you for this note, we have added this reference in the revision.

4. Add titles for Supplementary Tables.

Done, thanks.

5. Supplementary Fig. 1b: There is an error in figure label.

Changed, thanks.

6. Supplementary Fig. 3: Please provide E-value for each predicted motif.

Thanks for this comment, we have added them.

7. Supplementary Fig. 7c: Error in figure label.

Done, thank you.

Reviewer #2 (Remarks to the Author):

In this manuscript, Wang et al identify a new role of the Arabidopsis Telomere Repeat Binding proteins TRB1, TRB2 and TRB3 in the regulation of histone modification by triggering sequence-specific recruitment of the JMJ14 demethylase. Building on previous knowledge that TRBs can recruit PRC2 to telomeric motifs, the authors propose a model in which these proteins can, on one hand, trigger the removal of the active H3K4me3 chromatin mark, and on the other hand trigger the establishment of a repressive chromatin status through H3K27me3 deposition. That dual function is nicely supported by artificially targeting TRB1 to new loci upon protein engineering with a sequence-specific Zinc-Finger domain. The manuscript is very well written, nicely illustrated and should be of great interest for the readership of Nature Communications.

Thank you for the positive comments!

As detailed below, at several places it falls short in description of interpretation of the findings.

- I sincerely regret that no token was provided to the reviewers for a fair assessment of the NGS data quality

The token for the GEO data was included in the Reporting Summary during initial submission, and the title of this file was nb_nr-reporting-summary.pdf. We apologize that we didn't also add this information to the main text to make it clear for reviewers. The following is the detailed information for NGS data. All the high-throughput sequencing data generated in this study are accessible at NCBI's Gene Expression Omnibus (GEO) via GEO Series accession number GSE204681 (<https://www.ncbi.nlm.nih.gov/geo/query/acc.cgi?acc=GSE204681>). Enter token kduhcesuzrwztuv into the box.

We have added "All the high-throughput sequencing data generated in this study are accessible at NCBI's Gene Expression Omnibus (GEO) via GEO Series accession number GSE204681." (page 11, line 520-522)

When we re-analyzed the data during revision, we felt that the sequencing depth of our previous FLAG-TRB ChIP-seq was not deep enough, therefore, we re-sequenced it with more reads, which now has been added to the GEO as well. We have updated our results with the new TRB ChIP-seq data throughout the manuscript, and all the new results are either similar or

slightly cleaner than the old ones. Moreover, we also performed Myc-JMJ14 ChIP-seq in TRB-ZF backgrounds, and the sequencing data has been added as well.

- The notion that a single transcription factor like TRB1 can drive H3K4me3 erasure and H3K27me3 deposition is nicely defined in the general context of these two chromatin marks being generally mutually exclusive in the Arabidopsis epigenome. Yet, the proposed mechanism can only explain a small portion of this observation, as TRBs do not define the majority of PRC2 target genes. This aspect should be cautiously described. Ideally, it would be good to support the statement by an assessment of the proportion of loci at which TRB-JMJ14 association could trigger H3K4me3-to-H3K27me3 switches.

Thank you for pointing this out, and we agree with reviewer that the TRB1-JMJ14 connection can only explain a small portion of the mutual exclusion between H3K4me3 and H3K27me3, and we have deleted all the broad claims in the revision.

We have added “We demonstrate that TRB proteins not only recruit PRC2 complexes to deposit H3K27me3^{1,2}, but also recruit JMJ14 to remove H3K4me3, which can partially explain the mutual antagonism of these two histone marks over certain regions that are co-bound by JMJ14 and TRBs.” (Page 2, line 84-87) in the introduction.

We also have added, “It has been previously observed that H3K4me3 and H3K27me3 are in mutually exclusive domains of the Arabidopsis genome³; however, the underlying mechanism is not fully understood. Previous work in animals has shown that active histone marks such as H3K4me3 and H3K36me2/3 can inhibit the deposition of H3K27me3 to the same histone tail by PRC2^{4,5}. In this study, we found that Arabidopsis TRB proteins form a complex with PRC2 and JMJ14 complexes, and the mutation of TRBs not only impacted H3K27me3 deposition, but also H3K4me3 demethylation, suggesting that TRBs might act upstream of PRC2 and JMJ14 complexes and partially contribute to the mutual exclusion of H3K4me3 and H3K27me3 at certain regions that are co-bound by TRB1 and JMJ14.” (Page 8, line 332-340) in the discussion.

In addition, to provide a better visualization of the H3K27me3-H3K4me3 switch in the *trb1/2/3* mutant over TRB1-JMJ14 co-bound loci, we plotted H3K27me3 and H3K4me3 ChIP-seq signals in the *trb1/2/3* mutant versus Col-0 over the TRB1 Cluster 1 and Cluster 2 peaks, respectively. A very clear opposite trend of the changes between these two histone marks was only observed at the TRB1 Cluster1 peaks but not at the Cluster 2 peaks (Supplementary Fig. 7a, b).

To better estimate the proportion of TRB1-JMJ14 co-bound loci showing a switch between H3K4me3 and H3K27me3, we started with TRB1-JMJ14 co-bound genes instead of co-bound loci for this analysis, because the position of the peaks of H3K4me3 and H3K27me3 were not perfectly overlapping. Therefore, we generated a list of TRB1-JMJ14 co-bound genes with the normalized values of H3, H3K4me3, and H3K27me3 ChIP-seq signals in the *trb1/2/3* mutant and Col-0 plant (Supplementary Table 4). Among them, 860/2617 (33%) genes showed increased H3K4me3 signals and decreased H3K27me3 signals in the *trb1/2/3* mutant versus Col-0, suggesting that TRB1-JMJ14 triggers the switch from H3K4me3 to H3K27me3 in about 33% of their co-bound genes. This result has been added to Supplementary Table 4, and we have added “However, we observed a near perfect opposite trend of changes of H3K4me3 and H3K27me3 in the *trb1/2/3* mutant versus Col-0 at the TRB1 Cluster 1 peaks, we did not observe

this at Cluster 2 peaks (Fig. 3b and Supplementary Fig. 7a, b). Moreover, we found a total of 2617/3736 (70%) of JMJ14 bound genes overlapped with 2617/14882 (18%) of TRB bound genes (Supplementary Table 3), among which 860 (33%) showed a switch from H3K27me3 to H3K4me3 in the *trb1/2/3* mutant (Supplementary Table 4), suggesting that TRB1 and JMJ14 can trigger a shift from H3K4me3 to H3K27me3 in about a third of TRB1 and JMJ14 co-bound genes.” (Page 5, line 201-207).

- TRB1, 2 and 3 protein-protein association with JMJ14 is only supported by co-IP. This assay should be complemented by other approaches to determine whether interactions are direct (e.g. Y2H). They should also be confirmed to occur in planta (e.g., subnuclear co-localization), as co-IP positive results can result from biased associations occurring in cell extracts. Using a crosslinker does not necessarily help avoiding this.

As suggested by the reviewer, we performed both Y2H and BiFC experiments to confirm the interaction between TRBs, NAC052, and JMJ14. Our BiFC data showed clear interactions of TRBs with NAC052 and JMJ14 proteins, which has been added to Supplementary Fig. 1b, c. We have also added “We also fused JMJ14 and NAC052 with the N-terminal fragment of YFP, and TRBs with the C-terminal fragment of YFP to perform bimolecular fluorescence complementation (BiFC) in *Nicotiana benthamiana*. YFP signals were observed when TRBs were co-expressed with JMJ14 or NAC052, but not with the empty control vector (EV) (Supplementary Fig. 1b, c), suggesting that TRBs interact with both JMJ14 and NAC052.” (Page 3, line 114-118)

We did try Y2H as requested but, unluckily, the TRB proteins triggered a very strong self-activation effect, no matter whether they were fused with AD or BD and co-transformed with empty control vectors (EV). However, we did observe a stronger growth of yeast co-transformed with BD-TRBs and AD-JMJ14 or AD-NAC052, respectively, when compared with the yeasts co-transformed with BD-TRBs and AD-EV (please see the figures below), suggesting that TRB

might directly interact with JMJ14 and NAC052. However, we are hesitant to include this Y2H result in the revision due to the self-activation effect of TRB proteins.

Other than the BiFC result, we also provided multiple direct and indirect evidence to support the interaction between TRBs and JMJ14. 1). TRB IP-MS pulled down many peptides of JMJ14 and NAC052 (Supplementary Table 1); 2) Co-IP experiments in FLAG-TRBs x Myc-JMJ14 crossed lines showing the protein-protein interaction between TRBs and JMJ14 (Fig. 1a); 3). The Myc-JMJ14 ChIP-seq in the TRB-ZF x Myc-JMJ14 crossed lines showed an enrichment of myc-JMJ14 at many ZF binding sites, including *FWA* and several ZF off-target sites (Supplementary Fig. 17a-c); 4). TRBs and JMJ14 ChIP-seq also showed the co-localization of TRBs and JMJ14 over thousands of loci (Fig. 1b, c). 5). Motif prediction showed the ChIP-seq peaks of both JMJ14 and TRBs enriched at telobox TTAGGG and CTTGnnnnnCAAG motifs (Supplementary Fig. 4a, b). We hope these combined results address the reviewer's concern regarding the protein-protein interactions between TRBs and JMJ14.

- As shown in Figure S1b, JMJ14 activity with TRB1, 2, 3 is supported by the partial co-distribution of their chromatin association profiles along the genome. Yet, this analysis does not inform on whether the chromatin associations occur on the same loci in the same cells. This analysis could first be completed by stating the proportion of JMJ14 peaks that is linked to a TRB-associated gene and, vice versa, to which extent TRB1, 2, 3 peaks correspond to JMJ14-associated genes.

Thank you for this comment. Based on our ChIP-seq data, a total of 2617/3736 (70%) of JMJ14 bound genes overlapped with 2617/14882 (18%) TRB1 bound genes, suggesting that TRB1 peaks cover more than 70% JMJ14-associated genes, while JMJ14 peaks only cover 18% of TRB associated genes. We have added these results to Supplementary Table 3, and to the main text, "Moreover, we found a total of 2617/3736 (70%) of JMJ14 bound genes overlapped with 2617/14882 (18%) of TRB bound genes (Supplementary Table 3), among which 860 (33%) showed a switch from H3K27me3 to H3K4me3 in the *trb1/2/3* mutant (Supplementary Table 4), suggesting that TRB1 and JMJ14 can trigger a shift from H3K4me3 to H3K27me3 in about a third of TRB1 and JMJ14 co-bound genes." (Page 5, line 203-207).

-More importantly, given the observation that TRBs are often bound to chromatin independently of JMJ14, the latter correlation analysis should be completed by an experimental assessment of JMJ14 chromatin profile in *trb1,2,3* loss-of-function plants. This is a difficult but realistic experiment that can be achieved upon introgressing the FLAG-JMJ14 transgene in the available *trb* triple mutant plants in which one of the *trb* allele is in a heterozygous state. This would allow testing properly the proposed model by determining whether JMJ14 actually relies on TRBs for its recruitment at specific loci. Alternatively, the authors may complement the TRB1-ZF strategy by determining whether ectopic targeting of TRB1 to *FWA* and to all other off-targets results in systematic recruitment of JMJ14 to these loci.

Thank you for this excellent suggestion. We found that the rate at which we could find *trb1/2/3* homozygous mutants from *trb2/3* homo- and *trb1* heterozygous mutant was lower than 25%, which might be due to germination issues of the *trb1/2/3* mutant. In addition, the *trb1/2/3* mutants are extremely small as we showed in Supplementary Fig. 5. We sowed out a large number of *trb2/3* homo- and *trb1* heterozygous seeds (more than 500ml Eppendorf tube of seeds) on many MS plates and only obtained a small number of *trb1/2/3* homozygous mutants for histone ChIP-seq experiments. Therefore, it would be very difficult to collect enough plant materials with the FLAG-JMJ14 transgene in the *trb1/2/3* mutant for FLAG-JMJ14 ChIP-seq.

However, we were able to complete the second suggested experiment. We crossed Myc-JMJ14 and TRB-ZF together so we could perform Myc-JMJ14 ChIP-seq. As expected, we observed strong Myc-JMJ14 peaks over *FWA* in the TRB-ZF background, but not in the control background, suggesting that JMJ14 was indeed recruited by TRBs to *FWA* (Supplementary Fig. 17a). In addition, it was clear that Myc-JMJ14 was recruited to a subset of ZF off-targeting sites, which is consistent with our observation that endogenous TRB1 peaks only partially overlapped with endogenous JMJ14 peaks. We had found that endogenous JMJ14 was most frequently present at H3K4me3 depleted regions (Supplementary Fig. 17 a-c), probably due to its role in H3K4me3 demethylation. To determine whether the recruitment of Myc-JMJ14 to the ZF off-target sites is also correlated with pre-existing levels of H3K4me3, we plotted the normalized Myc-JMJ14 signals in TRB-ZFs versus control backgrounds over three H3K4me3 Clusters of ZF off-target sites corresponding to levels of pre-existing H3K4me3 (Cluster1 = high, Cluster2 = mid, Cluster3 = low). Consistent with the endogenous distribution of JMJ14 (Fig. 2b and c), Myc-JMJ14 were mainly recruited by TRB-ZFs to regions with medium or low levels of pre-existing H3K4me3 (Supplementary Fig. 16a, b). This data is also nicely consistent with our observation that the removal of H3K4me3 in TRB-ZF lines mainly occurred at regions with medium or low levels of H3K4me3 (Supplementary Fig. 16b).

We have added the following to the text “To determine whether the deposition of H3K27me3 or the removal of H3K4me3 were affected by pre-existing H3K27me3 or H3K4me3, respectively, ZF off-target sites were divided into three clusters according to pre-existing levels of either H3K4me3 or H3K27me3 in *fwa* plants (Cluster 1 = high, Cluster 2 = medium, and Cluster 3 = low). Intriguingly, TRB-ZFs triggered H3K27me3 deposition mainly at H3K27me3 Cluster 1 of ZF off-target sites (Supplementary Fig. 15a, b), suggesting that TRB-ZFs preferred to ectopically deposit H3K27me3 at ZF off-target sites already somewhat enriched with H3K27me3. Furthermore, we found that TRB-ZFs triggered H3K4me3 removal mainly at regions with medium (H3K4me3 Cluster 2) or low (H3K4me3 Cluster 3) levels of pre-existing H3K4me3 (Supplementary Fig. 16a, b). To investigate whether this was due to selective recruitment of JMJ14 by TRB-ZFs, Myc-JMJ14 was crossed with TRB-ZFs in order to perform Myc-JMJ14 ChIP-seq. As expected, a prominent Myc-JMJ14 peak appeared at *FWA* in TRB-ZFs, but not in the control (Supplementary Fig. 17a), confirming that TRBs recruit JMJ14 to *FWA*. Moreover, JMJ14 was mainly recruited to ZF off-target sites with medium or low levels of pre-existing H3K4me3 (Supplementary Fig. 17b, c), which is consistent with the result that TRB-ZFs triggered H3K4me3 removal mainly at H3K4me3 Cluster 2 and 3. Together with the data showing that TRB1 and JMJ14 co-localize to regions with low levels of H3K4me3 (Fig. 2b, c, and Supplementary Fig. 3c), these data suggest that TRBs can co-operate with JMJ14 to remove H3K4me2/3 at their co-bound sites, while TRBs fail to recruit JMJ14 to TRB1 unique binding sites enriched with H3K4me3.” (Page 7, line 284-302).

- TRBs have been shown to display the intrinsic property to associate to Telobox motifs, while JMJ14 associates with NAC050 and NAC052 that typically bind CTTGnnnnnCAAG motifs. The conclusion of TRBs and JMJ14 acting together theoretically implies that loci at which the TRB-JMJ14 mechanism operates display both a Telobox motif and a NAC050/2 binding motif. Is-this observed? If many exceptions are found, does-it mean that TRB-mediated JMJ14 recruitment to Telobox-containing genes acts independently of NAC050/2 proteins at loci bearing a Telobox but not a CTTGnnnnnCAAG motif? Investigating this aspect could allow addressing a central aspect of the findings, as to whether JMJ14 is recruited by TRBs to favor H3K27me3 as a main function, or whether

JMJ14 has other functions independently of TRBs relying on NAC050/2 but no TRB protein association.

Thank you for this very interesting suggestion. We find that both the Telobox (TTAGGG) and CTTGnnnnnCAAG motifs are enriched in the ChIP-seq peaks of TRBs, NAC050, NAC052, and MJM14 (Supplementary Fig. 4a, b). However, the top predicted motif bound by TRBs is TTAGGG, with the CTTGnnnnnGAAAG motif landing at rank number 61 (TRB1), 82 (TRB2) and 69 (TRB3). On the other hand, the predicted motifs for MJM14 were CTTGnnnnnCAAG at rank number 1, and TTAGGG at number 42. Similarly, the predicted motifs for the NACs were CTTGnnnnnCAAG at number 1, and TTAGGG at number 42 (NAC050) or 41 (NAC052) (Supplementary Fig. 4a, b). These results suggest that perhaps both mechanisms are at play, with MJM14 recruited to both TRB binding sites and NAC050/52 binding sites independently.

We also ran a specific analysis to address the reviewer's question of whether MJM14 might be recruited to TRB binding sites by specifically binding to both motifs simultaneously. We broke up TRB1 and MJM14 co-bound sites into four clusters based on whether or not they contain the exact Telobox binding sequence TTAGGG, the exact NAC binding sequence CTTGnnnnnCAAG, or both, or neither. We found that 12% of the peaks contained both motifs, 37% displayed neither motif, 37% only had Telobox motif, and 14% had only CTTGnnnnnCAAG motif (Supplementary Fig. 4c). Compared to shuffled peaks, the TRB/MJM14 co-bound sites showed an impressive enrichment for the CTTGnnnnnCAAG motif (Supplementary Fig. 4c). The segregation of peaks into these four buckets is of course very stringent since there will be some variation in these binding sequences at real binding sites, which probably explains why there are 37% of sequences that contain neither motif. Regardless, this analysis very nicely showed that the TRB ChIP-seq signals were higher at sites with the TTAGGG only, or with both sequences than sites with only CTTGnnnnnGAAG or neither, while MJM14 and the NACs bound more to the sites with only CTTGnnnnnGAAG or with both sequences, than to those with TTAGGG only or neither (Supplementary Fig. 4d). Again, this supports the conclusion that both mechanisms are at play, with MJM14 recruited to both TRB binding sites and NAC050/52 binding sites.

Finally, we plotted the changes of H3K4me3 in the *trb1/2/3* triple mutant or the *jmj14* mutant over these four clusters of sites and the results were very nice. In the *trb1/2/3* triple mutants, H3K4me3 was gained over the two clusters of regions that are lacking a CTTGnnnnnGAAG sequence (the Telobox only cluster and the neither motif cluster) but not at the two clusters that do have the CTTGnnnnnGAAG sequence. However, all four clusters of sites gain H3K4me3 in *jmj14-1*. This result suggests that MJM14-NAC likely can act at the CTTGnnnnnGAAG containing regions independently of TRBs. This result has been added as Fig. 3e.

We have added to the text, "Interestingly, both TTAGGG and CTTGnnnnnCAAG sequences were enriched in the TRBs, MJM14, and NAC050/052 ChIP-seq peaks (Supplementary Fig. 4a, b). However, the top predicted motif bound by TRBs was TTAGGG, with the CTTGnnnnnGAAAG motif ranking number 61 (TRB1), 82 (TRB2), or 69 (TRB3). On the other hand, the predicted motifs for MJM14, NAC050, and NAC052 bound sites were CTTGnnnnnCAAG at rank number 1, and TTAGGG at number 42 (MJM14), 42 (NAC050), or 41 (NAC052) (Supplementary Fig. 4a, b). These results suggest MJM14 may be independently recruited to telebox sites by TRBs and to CTTGnnnnnCAAG sites by NAC050/52.

We divided the TRB1 and MJM14 co-bound regions into four clusters based on whether they contained the Telobox binding sequence TTAGGG (37% of regions), the NAC binding sequence

CTTGnnnnnCAAG (14% of regions), or both sequences (12% of regions), or neither sequence (37% of regions) (Supplementary Fig. 4c). Compared to shuffled peaks, these TRB/JMJ14 co-bound sites showed a strong enrichment for the CTTGnnnnnCAAG sequence (Supplemental Fig. 4c). We found that the TRB ChIP-seq signals were higher at regions with the TTAGGG only sequence, or with both sequences, than at sites with only the CTTGnnnnnGAAG sequence or neither sequence, while JMJ14 and the NACs bound more to the regions with only CTTGnnnnnGAAG or with both sequences, than to those with TTAGGG only or neither sequence (Supplementary Fig. 4d). These results again support the conclusion that JMJ14 is likely recruited to both the TRB binding sites and the NAC050/052 binding sites independently.” (Page4, line 160-178).

We also added “We also plotted the changes of H3K4me3 in the *trb1/2/3* triple mutant or the *jmj14-1* mutant over the four clusters of TRB/JMJ14 co-bound sites described above (Supplementary Fig. 4 c, d). In the *trb1/2/3* triple mutant, H3K4me3 was gained over the two clusters of regions that are lacking a CTTGnnnnnGAAG sequence (the Telobox only cluster and the neither motif cluster) but not at the two clusters that do have the CTTGnnnnnGAAG sequence (Fig. 3e). However, all four clusters of sites gained H3K4me3 in *jmj14-1* (Fig. 3e) This result suggests that JMJ14-NAC050/052 likely can act at the CTTGnnnnnGAAG containing regions independently of TRBs, again suggesting that JMJ14 is likely recruited to both the TRB binding sites and the NAC050/052 binding sites independently.” (Page 5, line 213-221)

- A follow-up question is whether JMJ14 can associate with TRBs and to NAC050/2 in the same protein complex?

Our IP-MS, Co-IP, and BiFC data demonstrated that TRBs associated with JMJ14. TRB IP-MS also identified peptides of JMJ14, NAC050, and NAC052. We further performed BiFC and demonstrated that NAC052 also interacted with TRB proteins (Supplementary Fig. 1b, c). Although our Y2H data above showed a self-activation of TRB proteins, the co-transformation of TRBs with JMJ14 or NAC052 both showed a stronger growth of the yeast strains when compared with co-transformation of TRBs with the control vectors, suggesting that TRBs might directly interact with JMJ14 and NAC052. Furthermore, we re-analyzed the previous NAC050 and NAC052 ChIP-seq data from Xiaofeng Cao's paper¹⁹, and we found that NAC050 and NAC052 ChIP-seq signals were colocalized with almost all of the TRB1 and JMJ14 co-bound regions (TRB1 Cluster 1 peaks), but not with TRB1 unique peaks (TRB1 Cluster 2 peaks) (Fig. 1b and Supplementary Fig. 2b, c). Finally, we plotted JMJ14 and TRB1 ChIP-seq signals over NAC050 peaks, and the heatmap results showed that JMJ14 and TRBs ChIP-seq signals cover almost all the peaks of NAC050 (Supplementary Fig. 2b, c). All these data together suggest that JMJ14, TRBs, and NAC050/2 are interacting with each other in some fashion.

To answer definitely whether JMJ14 can associate with TRBs and to NAC050/2 simultaneously, we can think of a few ways of addressing this. One would be to do some sort of sequential IP-MS (like pulling on TRB, and then on JMJ14, and then looking for NAC050/2 peptides). We have never attempted something like this, and in our opinion, this would be very difficult to achieve given the losses associated with a second IP, together with the currently limited sensitive of MS. A second way would be to produce a structure of all of the relevant proteins together in a complex, which would be a huge undertaking. We hope the paper will be acceptable without these experiments.

- The model predicts that ectopic targeting of TRB1-ZF to FWA or to any other loci will

locally remove/reduce H3K4me3 and increase H3K7me3. This is what is shown in Figures 6, S10, S11 for a few loci where the model nicely works. Based on the authors' observations, is-it a general observation, and how does this compare when ZF off-targets do no, bear H3K4me3 and/or already bear H3K27me3 in *fwa* reference plants?

Thank you for this excellent suggestion. Indeed, as also noted above, the removal of H3K4me3 and the increase of H3K27me3 are only found at some ZF sites, and this correlates with varied levels of pre-existing epigenetic features. Specifically, we separated the ZF off target sites into three clusters according to the pre-existing levels of H3K4me3 or H3K27me3 in *fwa* plants, respectively. We then plotted the normalized H3K4me3 and H3K27me3 ChIP-seq signals of TRB-ZFs versus *fwa* over the clusters. We found that TRB-ZFs triggered removal of H3K4me3 mainly at the regions with medium (H3K4me3 Cluster 2) or low (H3K4me3 Cluster 3) levels of pre-existing H3K4me3 (Supplementary Fig. 16 a, b). Consistently, Myc-JMJ14 ChIP-seq in the Myc-JMJ14 plus TRB-ZF lines also showed that MJM14 was mainly recruited to H3K4me3 Cluster 2 and 3 sites (Supplementary Fig. 17a-c). Surprisingly, we found that H3K4me3 signal in TRB-ZFs was increased over the Cluster 1 ZF off-target sites (Supplementary Fig. 16 a, b). The possible explanations of this result could be 1). TRBs are not efficient at recruiting MJM14 to regions with high levels of H3K4me3, as the accessibility of MJM14 to these regions might be prohibited by unknown mechanisms (Supplementary Fig. 17c). 2). TRB IP-MS also pulled down several peptides of WDR5A (Supplementary Table1), which is an important component of the COMPASS-like complex associated with H3K4 methyltransferases. Furthermore, we also re-analyzed the WDR5A-ChIP-seq data from a previous paper (Wang et al., PNAS, 2022), which showed it was mainly located at the H3K4me3 enriched regions, and highly colocalized with TRB1 peaks, but not with MJM14 peaks (Supplementary Fig. 18b-d). These results suggest that TRB-ZF might recruit MJM14 to the H3K4me3 depleted regions for demethylation, but also recruit WDR5A and associated H3K4 methyltransferase activity to the H3K4me3 enriched regions to add methylation.

In addition, we also found that H3K27me3 was mainly deposited over the ZF off-target sites with high levels of pre-existing H3K27me3 (H3K27me3 Cluster 1) in TRB-ZF lines (Supplementary Fig. 15a and b), suggesting that TRB-ZF might selectively recruit the PRC2 complex to H3K27me3 enriched ZF off-target sites. This is consistent with the observation that TRBs only partially colocalized with the PRC2 complex and H3K27me3 over the genome.

We have added the following to the text “Notably, there were some ZF off-target sites that failed to show gene silencing, H3K4me3 demethylation, or H3K27me3 deposition in the TRB-ZFs T2 transgenic lines, which might be due to pre-existing epigenetic features of these sites.

To determine whether the deposition of H3K27me3 or the removal of H3K4me3 were affected by pre-existing H3K27me3 or H3K4me3, respectively, ZF off-target sites were divided into three clusters according to pre-existing levels of either H3K4me3 or H3K27me3 in *fwa* plants (Cluster 1 = high, Cluster 2 = medium, and Cluster 3 = low). Intriguingly, TRB-ZFs triggered H3K27me3 deposition mainly at H3K27me3 Cluster 1 of ZF off-target sites (Supplementary Fig. 15a, b), suggesting that TRB-ZFs preferred to ectopically deposit H3K27me3 at ZF off-target sites already somewhat enriched with H3K27me3. Furthermore, we found that TRB-ZFs triggered H3K4me3 removal mainly at regions with medium (H3K4me3 Cluster 2) or low (H3K4me3 Cluster 3) levels of pre-existing H3K4me3 (Supplementary Fig. 16a, b). To investigate whether this was due to selective recruitment of MJM14 by TRB-ZFs, Myc-JMJ14 was crossed with TRB-ZFs in order to perform Myc-JMJ14 ChIP-seq. As expected, a prominent Myc-JMJ14 peak appeared at *FWA* in TRB-ZFs, but not in the control (Supplementary Fig. 17a), confirming that TRBs recruit MJM14 to *FWA*. Moreover, MJM14 was mainly recruited to ZF off-target sites with medium or low

levels of pre-existing H3K4me3 (Supplementary Fig. 17b, c), which is consistent with the result that TRB-ZFs triggered H3K4me3 removal mainly at H3K4me3 Cluster 2 and 3. Together with the data showing that TRB1 and JMJ14 co-localize to regions with low levels of H3K4me3 (Fig. 2b, c, and Supplementary Fig. 3c), these data suggest that TRBs can co-operate with JMJ14 to remove H3K4me2/3 at their co-bound sites, while TRBs fail to recruit JMJ14 to TRB1 unique binding sites enriched with H3K4me3. Collectively, these results demonstrate that TRB-ZFs can target H3K27me3 deposition and H3K4me3 removal at many sites in the genome.

Surprisingly, we found that H3K4me3 signals in TRB-ZFs were increased over ZF off-target sites with high levels of pre-existing H3K4me3 (Supplementary Fig. 16 a, b). However, this phenomenon was not correlated with H3K27me3, because H3K27me3 was not altered at ZF off-target sites with high levels of H3K4me3 (Supplementary Fig. 18a). A possible explanation is that TRBs are not efficient at recruiting JMJ14 to the H3K4me3 enriched regions, as the accessibility of JMJ14 might be inhibited by unknown mechanisms. Furthermore, our TRB IP-MS pulled down several peptides of WDR5A protein (Supplementary Table 1), which is one of the major components of the COMPASS-like complex that associates with H3K4 methyltransferases^{20, 21}. We analyzed WDR5A-ChIP-seq data from a recent paper²², and we found that WDR5A was mainly located at the H3K4me3 enriched regions (Supplementary Fig. 18b, c), and highly colocalized with TRB1 but not JMJ14 (Supplementary Fig. 18b, d). Collectively, these results suggest that TRB-ZFs can recruit JMJ14 to regions with low or medium levels of pre-existing H3K4me3 to remove methylation, and in addition, TRB-ZFs might also recruit WDR5A and associated H3K4 methyltransferases to regions with high levels of pre-existing H3K4me3 to add methylation. However, additional investigation is needed to confirm the dual regulation of H3K4me3 by TRBs at varied loci." (Page 6-7, line 281-319)

- To support the claim of the new *trb1/2/3* mutant line having different phenotypes than previously published lines, the development of the different types of mutant lines should be compared upon growth in the same conditions.

We really appreciate the reviewer bringing up this subject. We contacted Dr. Franziska Turck about this issue and realize now that we had incorrectly assumed that her triple mutant was not a triple *trb1/2/3* null, but it turns out that it very likely is. In fact, she sent us photographs that show that her triple mutant is very similar to ours. We have revised the manuscript to drop the claim that our triple mutant is stronger. It is also not clear then why we saw a larger number of H3K27me3 DMRs than Franziska did, but this could be due to analysis issues, or because of plant growth differences. We have left this analysis in the paper because we used the output in our later analysis of the relationship between changes in H3K27me3 and H3K4me3 in the mutants.

- The rationale to conduct DNA methylation analyses in the *trb1/2/3* mutant line is obscure to me. How does this relate to H3K4me3 removal and JMJ14 function at transcribing genes?

The reason why we conducted DNA methylation analysis was initially because of our previously observed mild loss of non-CG DNA methylation in *jmj14* mutant over RdDM sites (Greenberg et al.; *Plos Genetics*, 2013). RdDM sites are often close to genes and so we thought we might see something. Therefore, we examined the DNA methylation level of the *trb1/2/3* mutant over RdDM sites, which also showed a reduction of non-CG DNA methylation (Fig. 5). As suggested by the reviewer, we also now plot the DNA methylation levels of the *trb1/2/3*

mutant vs Col-0 over TRB1 and JMJ14 associated genes. Here we also observed a mild reduction of CHH DNA methylation in the 5' and 3' ends of genes containing TRB1 and JMJ14 peaks in the *trb1/2/3* triple mutant when compared with Col-0 (Supplementary Fig. 10). This is likely because RdDM sites tend to be at the flanks of genes in euchromatin.

We have added to the text, "We also found that CHH DNA methylation was reduced in the *trb1/2/3* mutant in the 5' and 3' ends of genes containing TRB1 and JMJ14 peaks (Supplementary Fig. 10 a-c), which is likely because RdDM sites tend to be at the flanks of genes." (Page 6, line 243-245).

- The observation of CHH and CHG methylation loss in the *trb1/2/3* mutant line prompts the question as to whether TRB1-ZF expression in wild-type plants, which bear methylated *FWA*, can decrease the DNA methylation state of the *FWA* locus and influence flowering time.

In the original submission, TRB-ZF was introduced into the *fwa* background, where we found that *FWA* was silenced in a DNA methylation independent manner in TRB-ZF lines, even in plants showing an early flowering phenotype (Fig. 6a and b, and Supplementary Fig. 12a). To address the reviewer's question, we also transformed TRB-ZFs into the Col-0 background. McrBC-qPCR at *FWA* in these TRB-ZF lines indicated that there was no real difference of DNA methylation between TRB-ZFs and Col-0. In addition, the TRB-ZF lines in the Col-0 background did not flower later than Col-0 (please see figure below). Because these results are negative, we did not include them in the revision. We can add them if reviewer thinks these results are helpful.

-Last, the data indicate that TRB1/2/3 can also associate to domains stably enriched in H3K4me3. At these loci, TRBs most likely do not operate in H3K4me3 demethylation but may rather trigger H3K27me3 removal. This hypothesis is supported by the authors' MS data in which REF6 is one of the most abundant proteins co-purifying with TRB1, 2 and 3. Can the authors comment on this and potentially elaborate more in the conclusions?

Thank you for bringing out this point. We are hesitant to claim an interaction between REF6 and TRBs because we only observed REF6 peptides in the cross-linked IP-MS results, but not in the native TRB IP-MS data. Cross-link IP-MS can pull down many chromatin binding proteins, which do not necessarily associate with the tagged protein, because whole fragments of chromatin including nucleosomes are purified. In addition, the REF6 targeting motif (CTCTGYTY) was not observed in the predicted binding motifs of TRBs (Qiu et al., Nature Communication, 2019). Therefore, we are hesitant to include REF6 in the paper at all.

However, as the reviewer points out, TRBs did associate with H3K4me3 enriched regions, whereas TRB does not appear to operate in H3K4me3 demethylation (Supplementary Fig. 16 a and b), nor in H3K27me3 deposition (Supplementary Fig. 18a). Surprisingly, we found that H3K4me3 was increased at these ZF off-target sites with high levels of H3K4me3 by TRB-ZFs. This is probably due to the recruitment of WDR5A by TRBs, which is an important component of COMPASS-like complex that associates with H3K4 methyltransferases.

We have added to the manuscript “Surprisingly, we found that H3K4me3 signals in TRB-ZFs were increased over ZF off-target sites with high levels of pre-existing H3K4me3 (Supplementary Fig. 16 a, b). However, this phenomenon was not correlated with H3K27me3, because H3K27me3 was not altered at ZF off-target sites with high levels of H3K4me3 (Supplementary Fig. 18a). A possible explanation is that TRBs are not efficient at recruiting JMJ14 to the H3K4me3 enriched regions, as the accessibility of JMJ14 might be inhibited by unknown mechanisms. Furthermore, our TRB IP-MS pulled down several peptides of WDR5A protein (Supplementary Table 1), which is one of the major components of the COMPASS-like complex that associates with H3K4 methyltransferases^{20, 21}. We analyzed WDR5A-ChIP-seq data from a recent paper²², and we found that WDR5A was mainly located at the H3K4me3 enriched regions (Supplementary Fig. 18b, c), and highly colocalized with TRB1 but not JMJ14 (Supplementary Fig. 18b, d). Collectively, these results suggest that TRB-ZFs can recruit JMJ14 to regions with low or medium levels of pre-existing H3K4me3 to remove methylation, and in addition, TRB-ZFs might also recruit WDR5A and associated H3K4 methyltransferases to regions with high levels of pre-existing H3K4me3 to add methylation. However, additional investigation is needed to confirm the dual regulation of H3K4me3 by TRBs at varied loci.” (Page 7, line 304-319).

Minor points

- Lane 52, Arabidopsis has more than three TRB proteins, but only 3 have a clear coiled-coil domain.

Thank you for this comment, we have changed it to “Arabidopsis has three TRB proteins with a clear coiled-coil domain,”. (Page 2, line 52-53)

- Lane 54, TRB Myb-like domain has also been shown previously to enable TRB binding to telomeric motifs outside of telomeres

Thank you, we have changed it to “The Myb-like domain is required for DNA binding” (Page 2, line 54-55)

- Lane 62, "TRB1, 2, 3 are close homologs" is ambiguous. Is-it meant that they share a high sequence similarity or just that they functionally overlap?

We have changed it to “TRB1, 2, and 3 functionally overlapping.” in the revision. (Page 2, line 62)

- Lane 82, why mention TRB recruitment of PRC1/2 complexes? Is it meant PRC2 and/or LHP1?

Thank you for pointing this out, we should have just said PRC2. We have changed it to “We demonstrate that TRB proteins not only recruit PRC2 complexes to deposit H3K27me3^{1,2}, but also recruit JMJ14 to remove H3K4me3, which can partially explain the mutual antagonism of these two histone marks over certain regions that are co-bound by JMJ14 and TRBs.” (Page2, line 84-87).

- Figure S4 should compare trb1/2/3 mutant phenotypes to WT plants, especially because the genetics suggests that TRB1, 2 and 3 genes may have a dosage effect in trb1/2/3+/- plants.

- To support the claims at lane 154, figure S4 should provide information on the proper genotypes used, and their confirmation for each of the spotted individuals.

In the new Supplementary Fig.5, we grew a new set of the *trb1/2/3* homozygotes, *trb2/3* homozygotes, *trb2/3* homo- and *trb1* heterozygotes, as well as Col-0 wild type. The genotyping results of these plants were included. In addition, there was not very much phenotypic difference among the *trb2/3* homozygous, *trb2/3* homo- and *trb1* heterozygous mutants, and Col-0 plant (Supplementary Fig. 5).

Finally, thank you so much for looking in such great detail at the figures and helping us to improve them!

Reference

1. Zhou, Y., Hartwig, B., James, G.V., Schneeberger, K. & Turck, F. Complementary Activities of TELOMERE REPEAT BINDING Proteins and Polycomb Group Complexes in Transcriptional Regulation of Target Genes. *The Plant cell* **28**, 87-101 (2016).
2. Zhou, Y. et al. Telobox motifs recruit CLF/SWN-PRC2 for H3K27me3 deposition via TRB factors in Arabidopsis. *Nature genetics* **50**, 638-644 (2018).
3. Zhang, X., Bernatavichute, Y.V., Cokus, S., Pellegrini, M. & Jacobsen, S.E. Genome-wide analysis of mono-, di- and trimethylation of histone H3 lysine 4 in Arabidopsis thaliana. *Genome Biol* **10**, R62 (2009).
4. Schmitges, F.W. et al. Histone methylation by PRC2 is inhibited by active chromatin marks. *Molecular cell* **42**, 330-341 (2011).
5. Voigt, P. et al. Asymmetrically modified nucleosomes. *Cell* **151**, 181-193 (2012).
6. Yuan, L. et al. The transcriptional repressors VAL1 and VAL2 recruit PRC2 for genome-wide Polycomb silencing in Arabidopsis. *Nucleic acids research* **49**, 98-113 (2021).
7. Hecker, A. et al. The Arabidopsis GAGA-Binding Factor BASIC PENTACYSSTEINE6 Recruits the POLYCOMB-REPRESSIVE COMPLEX1 Component LIKE HETEROCHROMATIN PROTEIN1 to GAGA DNA Motifs. *Plant physiology* **168**, 1013-1024 (2015).

8. Xiao, J. et al. Cis and trans determinants of epigenetic silencing by Polycomb repressive complex 2 in Arabidopsis. *Nature genetics* **49**, 1546-1552 (2017).
9. Jeong, J.H. et al. Repression of FLOWERING LOCUS T chromatin by functionally redundant histone H3 lysine 4 demethylases in Arabidopsis. *PLoS one* **4**, e8033 (2009).
10. Yang, H. et al. Overexpression of a histone H3K4 demethylase, JMJ15, accelerates flowering time in Arabidopsis. *Plant cell reports* **31**, 1297-1308 (2012).
11. Liu, P. et al. The Histone H3K4 Demethylase JMJ16 Represses Leaf Senescence in Arabidopsis. *The Plant cell* **31**, 430-443 (2019).
12. Huang, S. et al. Arabidopsis histone H3K4 demethylase JMJ17 functions in dehydration stress response. *New Phytol* **223**, 1372-1387 (2019).
13. Yang, H. et al. A companion cell-dominant and developmentally regulated H3K4 demethylase controls flowering time in Arabidopsis via the repression of FLC expression. *PLoS genetics* **8**, e1002664 (2012).
14. Xiao, J., Lee, U.S. & Wagner, D. Tug of war: adding and removing histone lysine methylation in Arabidopsis. *Curr Opin Plant Biol* **34**, 41-53 (2016).
15. Liu, F. et al. The Arabidopsis RNA-binding protein FCA requires a lysine-specific demethylase 1 homolog to downregulate FLC. *Molecular cell* **28**, 398-407 (2007).
16. Jiang, D., Yang, W., He, Y. & Amasino, R.M. Arabidopsis relatives of the human lysine-specific Demethylase1 repress the expression of FWA and FLOWERING LOCUS C and thus promote the floral transition. *The Plant cell* **19**, 2975-2987 (2007).
17. Lu, F., Cui, X., Zhang, S., Liu, C. & Cao, X. JMJ14 is an H3K4 demethylase regulating flowering time in Arabidopsis. *Cell research* **20**, 387-390 (2010).
18. Ning, Y.Q. et al. Two novel NAC transcription factors regulate gene expression and flowering time by associating with the histone demethylase JMJ14. *Nucleic acids research* **43**, 1469-1484 (2015).
19. Zhang, S. et al. C-terminal domains of a histone demethylase interact with a pair of transcription factors and mediate specific chromatin association. *Cell discovery* **1** (2015).
20. Jiang, D., Kong, N.C., Gu, X., Li, Z. & He, Y. Arabidopsis COMPASS-like complexes mediate histone H3 lysine-4 trimethylation to control floral transition and plant development. *PLoS genetics* **7**, e1001330 (2011).
21. Jiang, D., Gu, X. & He, Y. Establishment of the winter-annual growth habit via FRIGIDA-mediated histone methylation at FLOWERING LOCUS C in Arabidopsis. *The Plant cell* **21**, 1733-1746 (2009).
22. Wang, Y. et al. The Arabidopsis DREAM complex antagonizes WDR5A to modulate histone H3K4me2/3 deposition for a subset of genome repression. *Proceedings of the National Academy of Sciences of the United States of America* **119**, e2206075119 (2022).

Reviewer #1 (Remarks to the Author):

It's good to see that FWA is also a target of JMJ14 and this targeting is mediated through TRB in the revised manuscript. I think the authors properly trimmed their claims such that their conclusions do not overstate their findings any longer. Although the revised manuscript does not expand the scope of the discovery, the authors addressed major portions of my concerns through the revision. I see that zinc-finger targeting to FWA is repetitively used in many works of this group. This might be an efficient and easy way for epigenome study. However, I would rather try more specific targeting to bona fide targets depending on baits and analyze targeting effects on them rather than off-targets, as this approach would be more logical, direct as well as biologically meaningful. We usually do not have clear idea on the epigenetic nature of those off-targets.

Reviewer #2 (Remarks to the Author):

In the revised version of the manuscript entitled "Arabidopsis TRB proteins function in H3K4me3 demethylation by recruiting JMJ14", the authors have satisfactorily addressed most of my concerns.

Yet, I got confused by Figure 3e that has been integrated in the revised version upon my request to test whether JMJ14 activity could differentially rely on TRBs for H3K4me3 de-methylation in different sequence contexts. While the meta-gene plots of Fig. 3e convincingly show that JMJ14-dependent H3K4me3 de-methylation relies on TRBs at telobox motifs, the corresponding heatmaps (lower panels) do not allow discerning any substantial changes between the 4 clusters in *jmj14-1* and *trb1/2/3* mutants. This discrepancy between the *trb1/2/3* meta-plots and heatmaps is surprising to me. One option is that differences seen in meta-gene plots could be driven by a few loci and not correspond to a general trend. Another option could be an error in the heatmaps presented.

In my opinion the conclusion of this analysis is important to support the proposed mechanistic model and I consequently think this point still requires additional feedback, complementary analyses or figure correction from the authors.

We would like to thank both reviewers again for the helpful comments on our revision. Here we are providing a point-by-point response to the remaining comments of both reviewers. **The reviewer comments are in bold type** and our response is in regular type. We hope the current response can fully address the reviewers' concerns.

REVIEWERS' COMMENTS

Reviewer #1 (Remarks to the Author):

It's good to see that FWA is also a target of JMJ14 and this targeting is mediated through TRB in the revised manuscript. I think the authors properly trimmed their claims such that their conclusions do not overstate their findings any longer. Although the revised manuscript does not expand the scope of the discovery, the authors addressed major portions of my concerns through the revision. I see that zinc-finger targeting to FWA is repetitively used in many works of this group. This might be an efficient and easy way for epigenome study. However, I would rather try more specific targeting to bona fide targets depending on baits and analyze targeting effects on them rather than off-targets, as this approach would be more logical, direct as well as biologically meaningful. We usually do not have clear idea on the epigenetic nature of those off-targets.

Thank you and are happy to see that the reviewer's concerns have been addressed in the revision.

With regard to the comment on off target sites, while we could have attempted to design zinc fingers to lots of other loci, this is a very difficult process, and we had to screen many different zinc fingers to discover the ZF108 that we have used in our work. In addition, even though it was designed for *FWA*, the ChIP-seq clearly shows its localization to the other promoters used in this study. So we call them "off targets" only because we didn't intend for them to go these targets, but they are nonetheless legitimate targets that can be studied. And with regard to the comment about not having a clear idea on the epigenetic nature of these off-targets, we would argue that we in fact have a very good idea, given that we have incorporated histone modification ChIP-seq, whole genome bisulfite sequencing, RNA-seq etc in this work.

Reviewer #2 (Remarks to the Author):

In the revised version of the manuscript entitled "Arabidopsis TRB proteins function in H3K4me3 demethylation by recruiting JMJ14", the authors have satisfactorily addressed most of my concerns.

Yet, I got confused by Figure 3e that has been integrated in the revised version upon my request to test whether JMJ14 activity could differentially rely on TRBs for H3K4me3 de-methylation in different sequence contexts. While the meta-gene plots of Fig. 3e convincingly show that JMJ14-dependent H3K4me3 de-methylation

relies on TRBs at telobox motifs, the corresponding heatmaps (lower panels) do not allow discerning any substantial changes between the 4 clusters in *jmj14-1* and *trb1/2/3* mutants. This discrepancy between the *trb1/2/3* meta-plots and heatmaps is surprising to me. One option is that differences seen in meta-gene plots could be driven by a few loci and not correspond to a general trend. Another option could be an error in the heatmaps presented.

In my opinion the conclusion of this analysis is important to support the proposed mechanistic model and I consequently think this point still requires additional feedback, complementary analyses or figure correction from the authors.

We apologize that the heatmaps were not as clear to the eye as the metaplots but they are derived from the same data, and they do show the same trend at the metaplots. We agree however that at a quick glance, the differences are not so clear and this is probably because of the scale and contrast used. We used a lower scale which makes the difference much clearer and provide an updated Figure 3e.

To address the option that the differences seen in meta-gene plots could be driven by a few loci and not correspond to a general trend, we sorted the 4 clusters based on the level of H3K4me3 increase in *trb1/2/3* mutant versus Col-0. The top 10% peaks of each cluster were removed and then used to redo the metaplots and heatmaps, which still showed a similar result (please see the figure below), suggesting that the difference seen in meta-gene plots is a general trend and not simply driven by a few loci.